# Heterogeneous levels of delta-like 4 within a multinucleated niche cell maintains muscle stem cell diversity

Susan Eliazer[1,2], Xuefeng Sun[1], Emilie Barruet[1,3], Andrew S Brack[1]*

[1]The Eli and Edythe Broad Center for Regenerative Medicine and Stem Cell Research, Department of Orthopedic Surgery, University of California San Francisco, San Francisco, United States; [2]Department of Biomedical Sciences, University of North Dakota School of Medicine and Health Sciences, Grand Forks, United States; [3]Departments of Surgery and Orofacial Sciences, Program in Craniofacial Biology, University of California San Francisco, San Francisco, United States

**Abstract** The quiescent muscle stem cell (QSC) pool is heterogeneous and generally characterized by the presence and levels of intrinsic myogenic transcription factors. Whether extrinsic factors maintain the diversity of states across the QSC pool remains unknown. The muscle fiber is a multinucleated syncytium that serves as a niche to QSCs, raising the possibility that the muscle fiber regulates the diversity of states across the QSC pool. Here, we show that the muscle fiber maintains a continuum of quiescent states, through a gradient of Notch ligand, Dll4, produced by the fiber and captured by QSCs. The abundance of Dll4 captured by the QSC correlates with the protein levels of the stem cell (SC) identity marker, Pax7. Niche-specific loss of Dll4 decreases QSC diversity and shifts the continuum to cell states that are biased toward more proliferative and committed fates. We reveal that fiber-derived Mindbomb1 (Mib1), an E3 ubiquitin ligase activates Dll4 and controls the heterogeneous levels of Dll4. In response to injury, with a Dll4-replenished niche, the normal continuum and diversity of the SC pool is restored, demonstrating bidirectionality within the SC continuum. Our data show that a post-translational mechanism controls heterogeneity of Notch ligands in a multinucleated niche cell to maintain a continuum of metastable states within the SC pool during tissue homeostasis.

*For correspondence:
Andrew.Brack@ucsf.edu

## Editor's evaluation

This is a strategic paper of relevance to both muscle and stem cell biologists. It bears on the generation of muscle stem cell diversity, and the factors bearing on this. Specifically, this paper identifies a particular, Notch ligand Dll4, as a myofiber-derived regulator of muscle stem cells.

## Introduction

Adult muscle stem cells (SCs) are essential for muscle tissue repair. Subsets of the SC pool are endowed with self-renewal potential and others are restricted to differentiation (*Rocheteau et al., 2012*; *Chakkalakal et al., 2014*; *Kuang et al., 2007*; *Scaramozza et al., 2019*; *Zammit et al., 2004*; *Olguin and Olwin, 2004*). This is consistent with a unidirectional and hierarchical relationship between SCs. Intrinsic and extrinsic cues regulate cell fate decisions of activated SCs to self-renew or differentiate (*Dumont et al., 2015*; *Kuang et al., 2008*). Due to low muscle tissue turnover, SCs exist predominantly in a quiescent state for the majority of adult life. Through single-cell RNA-sequencing (scRNA-seq) and transgenic reporter mice it is appreciated that the quiescent muscle stem cell (QSC)

pool is molecularly and phenotypically heterogeneous (*Dell'Orso et al., 2019*, *De Micheli et al., 2020*; *Chakkalakal et al., 2014*; *Rocheteau et al., 2012*; *Kuang et al., 2007*; *Kimmel et al., 2020*). These different cell states enable SCs to exhibit functional heterogeneity in response to activation and injury cues. It is not known how these diverse states are maintained across the QSC pool. SC quiescence is actively maintained by paracrine-acting cues from the muscle fiber, serving as a niche cell (*Bischoff, 1990*; *Goel et al., 2017*; *Eliazer et al., 2019*). In contrast to niche cells across many SC compartments, each muscle fiber is a multinucleated syncytium, that exhibits transcriptional diversity across the myonuclei to provide spatial control for specialized functions (*Kim et al., 2020*; *Petrany et al., 2020*). Does the multinucleated niche cell regulate the diversity of states across the QSC pool?

## Results

### Adult QSCs exist in a continuum of molecular cell states

We first asked whether the QSC pool is composed of a continuum of cell states during tissue homeostasis. We stained freshly isolated single muscle fibers from adult mice, with the SC identify marker, Pax7 (*Sambasivan et al., 2011*; *von Maltzahn et al., 2013*; *Lepper et al., 2011*) and Ddx6 (p54/RCK), an RNA helicase found enriched in P-bodies and stress granules (*Buchan and Parker, 2009*). Pax7 and Ddx6 are expressed in QSCs and decrease during activation and commitment (*Crist et al., 2012*; *Zammit et al., 2006*). A density map of Pax7 and Ddx6 expression shows a broad range of expression levels across the QSC pool (*Figure 1A–C*). A bivariate plot shows a positive correlation between Pax7 and Ddx6 (*Figure 1D*). Therefore, based on the expression profile of two different markers, the QSC pool is composed of a continuum of molecular states at tissue homeostasis. This supports models inferred from scRNA-seq analysis on QSCs (*De Micheli et al., 2020*; *Dell'Orso et al., 2019*, *Kimmel et al., 2020*).

To determine the fate bias of cells along the continuum, we isolated Pax7$^{high}$-, Pax7$^{medium}$-, and Pax7$^{low}$-expressing populations of SCs from *Pax7-nGFP* reporter mice (*Rocheteau et al., 2012*; *Figure 1—figure supplement 1A–C*), and analyzed cell cycle entry and differentiation. As expected, the Pax7$^{low}$ population entered cell cycle and differentiated faster than the Pax7$^{high}$ expressers, suggesting that Pax7$^{high}$ SCs are in a more dormant state and the Pax7$^{low}$ SCs exist in a primed state (*Rocheteau et al., 2012*). We find that the Pax7$^{medium}$ population is in an intermediate molecular and phenotypic state between the Pax7$^{high}$ and Pax7$^{low}$ QSCs (*Figure 1—figure supplement 1D, E*). Therefore, the QSC pool exists in a continuum of cell states that gives rise to a continuum of fates.

### Heterogeneous expression of Dll4 in a multinucleated niche cell is coupled to QSC diversity

The multinucleated muscle fiber functions as a niche cell for the SC, regulating the depth of quiescence and rate of activation in response to injury (*Eliazer et al., 2019*). The Delta-Notch signaling pathway is an evolutionarily conserved intercellular signaling pathway for cell fate specification (*Kopan and Ilagan, 2009*). Adult muscle QSCs display active notch signaling (*Bjornson et al., 2012*; *Mourikis et al., 2012*; *Fukada et al., 2011*; *Low et al., 2018*; *Verma et al., 2018*). Overexpression of NICD1 increases Pax7 expression (*Wen et al., 2012*) and Rbpj, a transcriptional coactivator of the notch signaling pathway, maintains the SC pool by repressing differentiation (*Bjornson et al., 2012*; *Mourikis et al., 2012*). A transgenic notch reporter mouse line where green fluorescent protein (GFP) is under the control of C promoter-binding factor 1 (CBF1) (*Duncan et al., 2005*) exhibits variable levels of notch activity in QSCs (*Bjornson et al., 2012*). Using the same reporter, we find that notch activity levels exist on a continuum across the SC pool and positively correlates with Pax7 expression levels (*Figure 1E, F*), raising the possibility that Notch ligands maintain the continuum of diverse cell states across the QSC pool.

To identify notch ligands specifically expressed in adult myofibers, we first examined Notch ligands on isolated single muscle fibers from postnatal day 3 (p3), postnatal day 7 (p7), and adult through microarray analysis. *Dll4* transcripts in the fiber showed the most enrichment as the SC transitioned from proliferating myogenic progenitors (postnatal) to becoming quiescent (adult) (*Figure 2—figure supplement 1A*). RT-qPCR(Real Time-quantiative PCR) on isolated adult single muscle fibers show the expression of *Dll4* transcripts, which was confirmed by RNAscope and immunofluorescence (*Figure 2—figure supplement 1B, C* and *Figure 2A*; *Kann and Krauss, 2019*). *Dll4* transcripts are

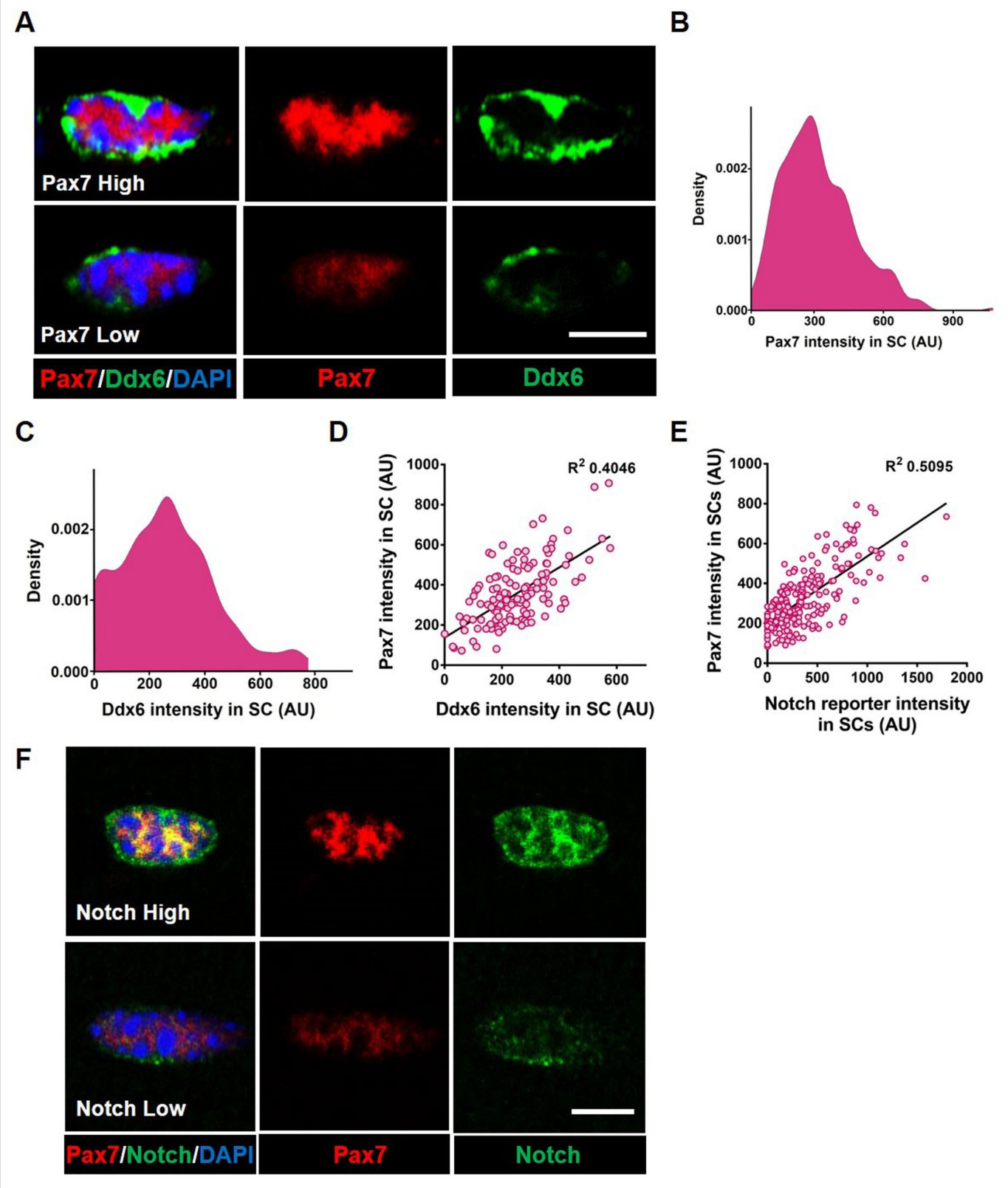

**Figure 1.** Adult quiescent muscle stem cells (QSCs) exist in diverse cell states. (**A**) Representative images of a high and low Pax7+ stem cell (SC) on a freshly isolated muscle fiber that has corresponding high and low levels of Ddx6 protein. (**B**) A density map of Pax7 intensity in QSCs (*n* = 3). (**C**) A density map of Ddx6 intensity in QSCs (*n* = 3). (**D**) A bivariate plot between Pax7 and Ddx6 intensity in QSCs (*n* = 2). (**E**) A bivariate plot between Pax7

Figure 1 continued

and Notch reporter intensity in QSCs (*n* = 3). (**F**) Image of SC with high Pax7 expression that expresses high Notch activity and an SC with low Pax7 intensity displaying low Notch activity. Scale bars, 5 µm in (**A**) and (**F**).

The online version of this article includes the following source data and figure supplement(s) for figure 1:

**Source data 1.** Adult QSCs exist in diverse cell states.

**Source data 2.** Adult QSCs exist in diverse cell states.

**Source data 3.** Adult QSCs exist in diverse cell states.

**Source data 4.** Adult QSCs exist in diverse cell states.

**Source data 5.** Adult QSCs exist in diverse cell states.

**Source data 6.** Adult QSCs exist in diverse cell states.

**Source data 7.** Adult QSCs exist in diverse cell states.

**Figure supplement 1.** Adult quiescent muscle stem cells (QSCs) exist in a continuum of cell states that give rise to a continuum of fates when activated.

expressed in less than 1% of QSCs (*Kimmel et al., 2020*; *De Micheli et al., 2020*), suggesting that the muscle fiber is a source of SC-bound Dll4. Since the QSC pool is distributed as a continuum of cell states, we hypothesized that Dll4 expression across the multinucleated niche cell is heterogeneous. Single muscle fibers stained with anti-Dll4 show that Dll4 protein formed clusters along the muscle fiber and were enriched around the QSCs. Further analysis revealed that heterogeneous spatial expression of Dll4 protein along the muscle fiber correlated with the amount of Dll4 captured by the SCs (*Figure 2A*, *Figure 2—figure supplement 2*). We observed a positive correlation between the amount of Dll4 foci on the fibers and the intensity of Dll4 present on the adjacent QSC (*Figure 2B*).

To directly link Dll4 localization with SC diversity, we isolated and stained SCs for Dll4 and Pax7. A bivariate plot of Pax7 and Dll4 expression within SCs reveals a positive correlation between the Dll4 captured by the SC and Pax7 intensity (*Figure 2C*). Comparison between Dll4 protein and Pax7 levels using a Pax7-nGFP reporter, confirmed the positive correlation between Pax7 and Dll4 protein expression (*Figure 2D, E*). In contrast, the localization of *Dll4* transcript on the muscle fiber did not correlate with Pax7 expression in the adjacent SC (*Figure 2—figure supplement 1C and D*).

We next asked about the distribution of Dll4 protein along muscle fibers. Quantification of Dll4 expression levels on single Extensor Digitorum Longus (EDL) muscle fibers showed that Dll4 expression is variable along the fiber, and not linked to the neuromuscular junction based on α-bungarotoxin expression or the myotendinous junction based on location (*Figure 2—figure supplement 2*). We find no consistent pattern of Dll4 protein along the fibers. Therefore, Dll4 spatial distribution does not map to these known anatomically defined regions of freshly isolated single muscle fibers. In addition, fewer than 10% of fibers are exclusively Dll4$^{High}$ or Dll4$^{low}$, suggesting SC diversity is coordinated in a fiber-autonomous manner. Dll4 expression was also variable in regions devoid of SCs, suggesting the presence of an SC does not dictate Dll4 levels along muscle fibers. Instead, the intensity of Dll4 captured by SCs and the Pax7 expression correlated with the level of Dll4 expression along the muscle fiber (*Figure 2—figure supplement 2*). Therefore, spatial heterogeneity and levels of niche-derived Dll4 are coupled with a continuum of diverse molecular SC states that are biased to different phenotypic fates.

## Niche-derived Dll4 maintains SC diversity

To determine if Dll4 from the muscle fiber controlled SC diversity, we deleted *Dll4* (*Hozumi et al., 2008*) specifically in the adult muscle fibers using a tamoxifen (tmx)-inducible human ACTA-*CreMer* mouse line (*McCarthy et al., 2012*) (transgenic mice are herein called as MF-*Dll4$^{fl/fl}$*) (*Figure 3A*). The abundance of Dll4 in the muscle fiber and captured by SCs decreased in MF-*Dll4$^{fl/fl}$* compared to control, suggesting that the muscle fiber is the major source of SC-bound Dll4 in uninjured muscle (*Figure 3B–D*). It is possible that cell sources other than the muscle fiber could provide Dll4 to SCs in other contexts (*Verma et al., 2018*). We observed that Pax7 and Ddx6 expression in SCs from MF-*Dll4$^{fl/fl}$* fibers decreased compared to controls (*Figure 3E, G*). Density maps reveal a shift in the distribution of expression levels of Pax7 and Ddx6. We calculated the variance of the expression levels to gauge the spread of the distribution. The decrease in variance in MF-*Dll4$^{fl/fl}$* compared to controls, indicates reduced diversity across the QSC pool, effectively reducing the range of states across the

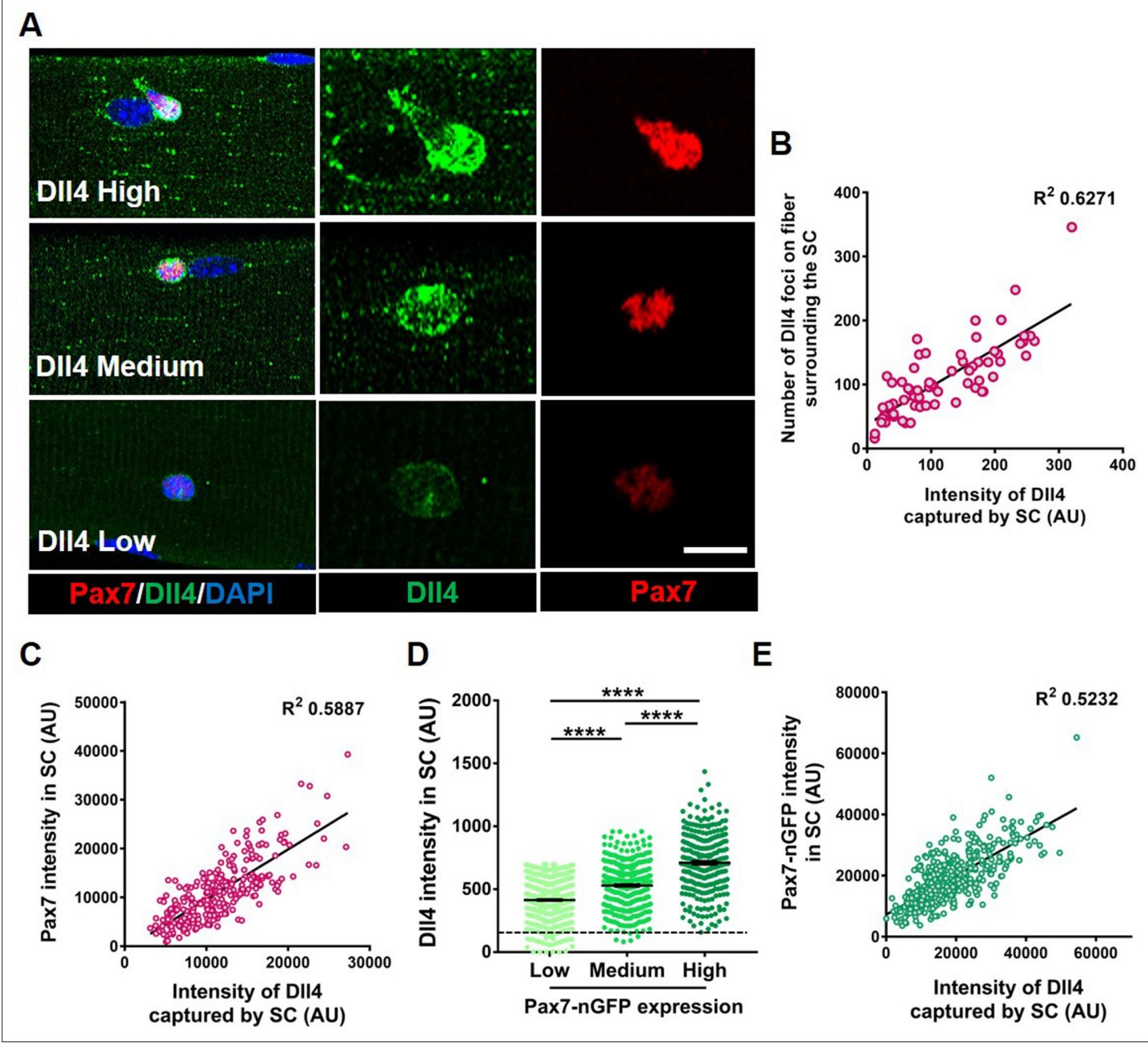

**Figure 2.** Heterogeneous expression of Dll4 on the muscle fiber correlates with stem cell (SC) diversity. (**A**) Representative images of regions of muscle fiber expressing heterogeneous levels of Dll4 protein, which corresponds to the amount of Dll4 captured by the adjacent SC and the expression of Pax7. (**B**) A bivariate plot between the number of Dll4 foci on fiber adjacent to an SC and the intensity of Dll4 captured by the SC ($n > 3$ mice). (**C**) An XY plot on isolated wildtype quiescent muscle stem cells (QSCs) shows positive correlation between intensity of Dll4 captured by SC and its Pax7 intensity ($n = 3$ mice). (**D**) Freshly isolated green fluorescent protein (GFP) low, medium, and high expressing SCs from transgenic *Pax7-nGFP* reporter mice were stained for Dll4 ($n = 3$). (**E**) An XY plot shows positive correlation between intensity of Dll4 captured by SC and its GFP (Pax7) intensity ($n = 3$ mice). Error bars, mean ± standard error of the mean (SEM); ****$p < 0.0001$. Scale bars, 10 µm in (**A**).

The online version of this article includes the following source data and figure supplement(s) for figure 2:

**Source data 1.** Heterogeneous expression of Dll4 on the muscle fiber correlates with SC diversity.

**Source data 2.** Heterogeneous expression of Dll4 on the muscle fiber correlates with SC diversity.

**Source data 3.** Heterogeneous expression of Dll4 on the muscle fiber correlates with SC diversity.

**Source data 4.** Heterogeneous expression of Dll4 on the muscle fiber correlates with SC diversity.

*Figure 2 continued*

**Source data 5.** Heterogeneous expression of Dll4 on the muscle fiber correlates with SC diversity.

**Source data 6.** Heterogeneous expression of Dll4 on the muscle fiber correlates with SC diversity.

**Source data 7.** Heterogeneous expression of Dll4 on the muscle fiber correlates with SC diversity.

**Source data 8.** Heterogeneous expression of Dll4 on the muscle fiber correlates with SC diversity.

**Figure supplement 1.** *Dll4* transcripts are highly expressed in adult muscle fibers.

**Figure supplement 2.** Spatial distribution of Dll4 along muscle fibers does not map to known anatomical locations.

continuum (*Figure 3F, H*). We observed a 50% decrease in the total number of Pax7$^+$ SCs in MF-*Dll4*$^{fl/fl}$ fibers compared to controls (*Figure 3—figure supplement 1A, B*), suggesting a SC loss phenotype.

Deletion of *Rbpj* from SCs reduced SC population by over 95% (*Bjornson et al., 2012*; *Mourikis et al., 2012*). We wondered whether the partial SC ablation after *Dll4* deletion was due to a more modest effect on notch signaling. To determine the contribution of niche-derived Dll4 to notch activity in SCs, we crossed mice harboring the notch reporter *CBF1-GFP* (*Duncan et al., 2005*) with MF-*Dll4*$^{fl/fl}$. We observed a 95% decrease in notch reporter activity after *Dll4* deletion from the niche (*Figure 3I*), suggesting that Dll4 from the fibers nonautonomously regulates Notch activity in the QSCs and that Dll4 from the muscle fiber is the dominant source of notch ligand. This argues against differential notch activity to explain the differences in SC loss phenotypes.

A decrease in SC diversity, number and Pax7 levels can be explained by a loss of Pax7$^{high}$ SCs (due to apoptosis or fusion), or a shift in the continuum toward a more committed fusion-competent state, followed by fusion of SCs expressing the lowest levels of Pax7. qRT-PCR on QSCs from control and MF-*Dll4*$^{fl/fl}$ revealed a decrease in *Pax7* (SC marker) and an increase in *Myod* (activation marker) and *Myog* (differentiation marker) transcripts (Figure 5B). The simultaneous loss of *Pax7* and induction of *Myog* a gene not normally expressed in QSCs suggests a shift in the QSC continuum. While a loss of Pax7$^{high}$ cells alone would not increase *Myog* expression, we cannot exclude a loss of Pax7$^{high}$ SCs in addition to the shift in continuum.

To resolve these possibilities, rather than deleting, we reduced *Dll4* levels in the muscle fiber by injecting control and MF-*Dll4*$^{fl/fl}$ mice submaximal doses of tmx (75 mg/kg/day for seven consecutive days). Reducing *Dll4* in fibers redistributed the SC population along the continuum toward lower Pax7 levels without decreasing their numbers (*Figure 4*). This suggests that heterogeneous levels of Dll4 in muscle fibers maintain a distribution of metastable states across the QSC pool during tissue homeostasis.

Deletion of *Dll4* in Pax7$^+$ SCs using a *Pax7-CreER* transgenic mouse line (*Nishijo et al., 2009*) (referred to as SC-*Dll4*$^{fl/fl}$) did not change Pax7$^+$ SC number, compared to controls (*Figure 3—figure supplement 2*), suggesting that Dll4 does not have an autocrine role in QSCs. Together, these results suggest that Dll4 from the muscle fiber is required to maintain the continuum of diverse cellular states across the QSC pool.

## Dll4 from the muscle fiber constrains the proliferative and commitment potential across the QSC pool

To analyze the phenotypic fates of the QSCs that remained in a Dll4-depleted niche, we cultured isolated single muscle fibers from control and MF-*Dll4*$^{fl/fl}$ mice and analyzed them at 0, 30, and 48 hr (*Figure 5A*). Quantification of SC numbers per fiber reveal that the differences observed between control and MF-*Dll4*$^{fl/fl}$ becomes narrow over time in culture (*Figure 5C*, *Figure 5—figure supplement 1A*). This observation is consistent with the result that SCs on a Dll4-depleted niche entered cell cycle faster than controls (*Figure 5D*, *Figure 5—figure supplement 1B*). The absolute number of Myogenin$^+$ cells per fiber at 48 hr was increased, suggesting that a fraction of the remaining SCs on a Dll4-depleted niche rapidly entered the differentiation program when exposed to mitogen (*Figure 5E*). These data are consistent with a redistribution of the QSC pool along the continuum toward lower levels of Pax7 (*Figure 1—figure supplement 1*). Induction of *Myogenin* is consistent with the antidifferentiative role of Rbpj (*Bjornson et al., 2012*; *Mourikis et al., 2012*). Based on prior work, we would not have predicted an increase in SC proliferation in mitogen after niche depletion of Dll4 (*Bjornson et al., 2012*; *Mourikis et al., 2012*). This likely reflects the relative fraction of SCs that remain after *Rbpj* deletion in SCs versus *Dll4* deletion from the niche.

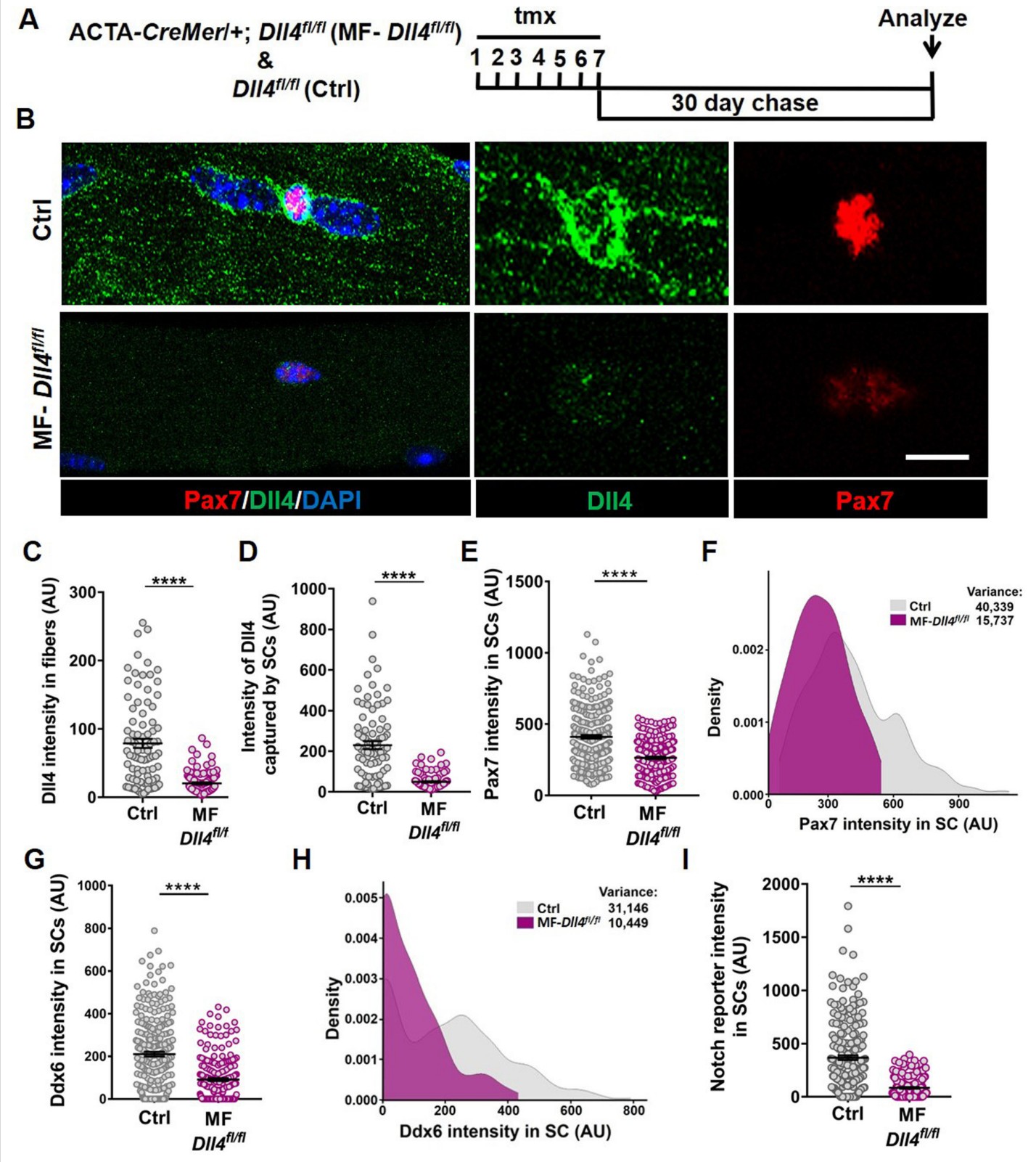

**Figure 3.** Muscle fiber-derived Dll4 maintains a continuum of diverse states in the quiescent muscle stem cell (QSC) pool. (**A**) Schematic representation of the experimental design. (**B–F**) Representative images of Dll4 and Pax7 expression in Control and MF-*Dll4*<sup>fl/fl</sup> fibers and stem cells (SCs) (in B), quantification of Dll4 intensity in fibers (in C), intensity of Dll4 captured by SCs (in D), Pax7 intensity in SCs (in E), and a density map of Pax7 intensity in SCs (in F) (*n* = 3 mice). (**G, H**) Ddx6 intensity in SCs (in G) and density map of Ddx6 intensity in SCs (in H) on Control and MF-*Dll4*<sup>fl/fl</sup> fibers (*n* = 3 mice).

*Figure 3 continued on next page*

*Figure 3 continued*

(**I**) Notch reporter intensity levels in Control and MF-*Dll4*<sup>fl/fl</sup> SCs (*n* = 3). Error bars, mean ± standard error of the mean (SEM); \*\*\*\*p < 0.0001; scale bars, 10 μm in (**B**).

The online version of this article includes the following source data and figure supplement(s) for figure 3:

**Source data 1.** Muscle fiber derived Dll4 maintains a continuum of diverse states in the QSC pool.

**Source data 2.** Muscle fiber derived Dll4 maintains a continuum of diverse states in the QSC pool.

**Source data 3.** Muscle fiber derived Dll4 maintains a continuum of diverse states in the QSC pool.

**Source data 4.** Muscle fiber derived Dll4 maintains a continuum of diverse states in the QSC pool.

**Source data 5.** Muscle fiber derived Dll4 maintains a continuum of diverse states in the QSC pool.

**Source data 6.** Muscle fiber derived Dll4 maintains a continuum of diverse states in the QSC pool.

**Source data 7.** Muscle fiber derived Dll4 maintains a continuum of diverse states in the QSC pool.

**Figure supplement 1.** Deletion of *Dll4* in the niche causes a reduction in the number of stem cell (SC).

**Figure supplement 2.** Dll4 does not have a cell autonomous role in quiescent muscle stem cells (QSCs).

We next examined the effect of a Dll4-depleted niche on fate potential of the remaining SCs in response to muscle injury. Using the tmx-inducible MF-*CreMer/+* genetic model, the niche is genetically modified, but the SCs are genetically wildtype. The wildtype SCs reform the adult muscle fibers during regeneration (*Eliazer et al., 2019*). Therefore, any regenerative phenotype observed is directly due to the loss of Dll4 in the niche prior to injury. Fourteen days after injury, muscle fiber size in MF-*Dll4*<sup>fl/fl</sup> TA muscle was significantly smaller than controls (*Figure 5F, G*), consistent with the rapid entrance of some SCs into the differentiation program, as reported after deletion of *Rbpj* in SCs (*Bjornson et al., 2012*; *Mourikis et al., 2012*). The impaired differentiation phenotype in MF-*Dll4*<sup>fl/fl</sup> TA muscle is also seen 40 days after injury (*Figure 5—figure supplement 2*). The inability to repair muscle fibers after a loss in SC diversity suggests a reduced pool of fusion-competent progenitors.

We next asked whether the remaining population of Pax7+ SCs in a Dll4-depleted niche could reestablish the normal continuum of Pax7+ states in the injury model. In contrast to the contralateral uninjured muscle, the average Pax7 levels and the variance across the SC population was not different in a Dll4-replenished niche and control niche (*Figure 5H, I*). In addition, the number of Pax7+ cells was similar to control regenerated muscle (*Figure 5J*). This was unanticipated due to the decrease in Pax7 levels and a shift along the continuum toward a more committed state in the QSC pool (*Rocheteau et al., 2012*; *Kuang et al., 2007*). Therefore, SCs expressing modest levels of Pax7 have the potential to produce SCs with higher levels of Pax7, revealing an unappreciated level of plasticity and bidirectionality along the SC continuum during muscle regeneration.

## Mib1 directs spatial patterning and activation of Dll4

Single nucleus RNA sequencing has uncovered substantial transcriptional diversity across muscle fibers (*Kim et al., 2020*; *Petrany et al., 2020*). However, Dll4 activity is regulated post-translationally. Mindbomb1 (Mib1), is an E3 ubiquitin ligase that activates all Notch ligands in the signal sending cell by adding mono-ubiquitin groups onto the ligands (*Koo et al., 2005*; *Koo, 2007*). Mib1 in the retinal pigment epithelium (RPE) activates Notch signaling in the adjacent retinal progenitor cell (RPC) by localizing active Notch ligands in the RPE–RPC contacts (*Ha et al., 2017*). The hypothalamus–pituitary–gonadal axis has been shown to induce Mib1 in muscle fibers during development, thus activating notch signaling in juvenile SCs (*Kim et al., 2016*). To determine if the heterogeneous pattern of Dll4 on the adult muscle fiber is directed by Mib1, we analyzed the spatial expression of Mib1 on adult muscle fibers. We stained isolated single muscle fibers with Mib1 antibody and found heterogeneous expression pattern of Mib1 on individual muscle fibers. Similar to Dll4 expression, we observed a positive correlation between the intensity of Mib1 expression on muscle fibers and Pax7 intensity within the adjacent SC (*Figure 6A, B*). Formation of Dll4 clusters is a marker of the activated form of the ligand (*Le Borgne and Schweisguth, 2003*). To test whether Mib1 played a role in clustering and activating Dll4 on the muscle fibers, *Mib1* floxed mice (*Koo, 2007*) were crossed with tamoxifen-inducible human *ACTA1-CreMer* mice (transgenic: hereafter called as MF-*Mib1*<sup>fl/fl</sup>). After a 30-day chase, the expression of *Mib1* transcripts was decreased in the Mib1-depleted myofibers compared to controls (*Figure 7A, B*). Analysis of Dll4 expression in control and MF-*Mib1*<sup>fl/fl</sup> isolated single muscle

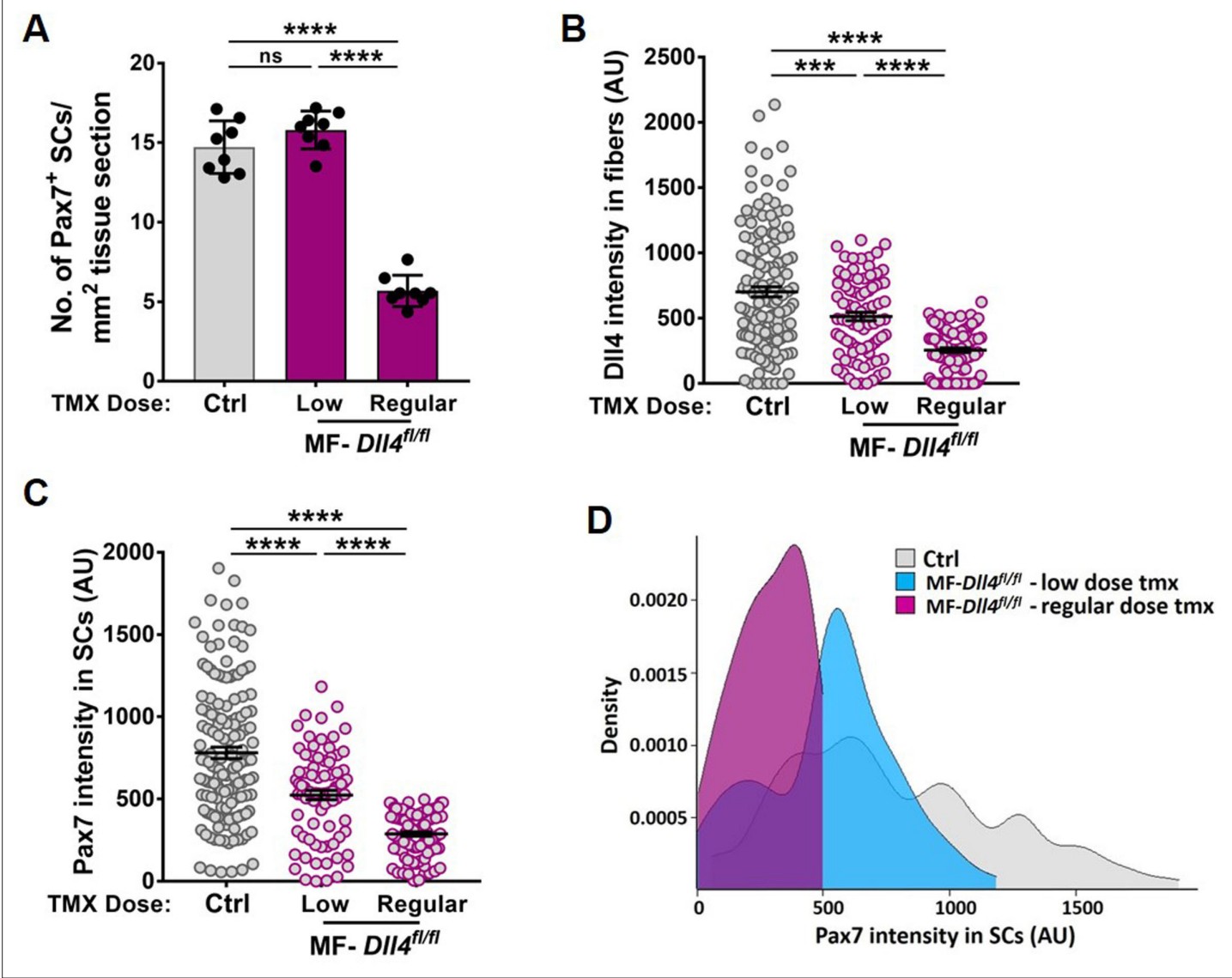

**Figure 4.** Reduction of *Dll4* in the niche causes a shift in the continuum toward lower Pax7 levels. (A–C) Quantification of the number of Pax7⁺ stem cells (SCs) (in A), Dll4 intensity in muscle fibers (in B), Pax7 intensity in SCs (in C) of Control mice compared to MF-*Dll4^fl/fl* mice that were given low dose (75 mg/kg/day) or regular dose (150 mg/kg/day) of tmx to reduce or completely ablate the expression of Dll4, respectively. (D) Density plot of the Pax7 intensity in SCs of Control mice compared to MF-*Dll4^fl/fl* mice that were given the two different doses of tmx (*n* = 3 mice). Error bars, mean ± standard error of the mean (SEM); ns, nonsignificant, ***p < 0.001, ****p < 0.0001.

The online version of this article includes the following source data for figure 4:

**Source data 1.** Reduction of Dll4 in the niche causes a shift in the continuum towards lower Pax7 levels.

**Source data 2.** Reduction of Dll4 in the niche causes a shift in the continuum towards lower Pax7 levels.

fibers, revealed a loss of Dll4 clusters and a reduction in the intensity of Dll4 expression in Mib1-depleted niche (*Figure 6C–E*). Therefore, Mib1 directs the activation and heterogeneous patterning of Dll4 within a multinucleated niche cell.

## Niche-derived Mib1 maintains a continuum of states across the QSC pool

Based on the loss of SC-bound Dll4 after Mib1 deletion in the niche, we analyzed the levels and diversity of Pax7 across the QSC pool on a Mib1-depleted niche. Compared to controls, Pax7 levels decreased, the variance across the QSC pool was reduced and SC number was less in a Mib1-depleted

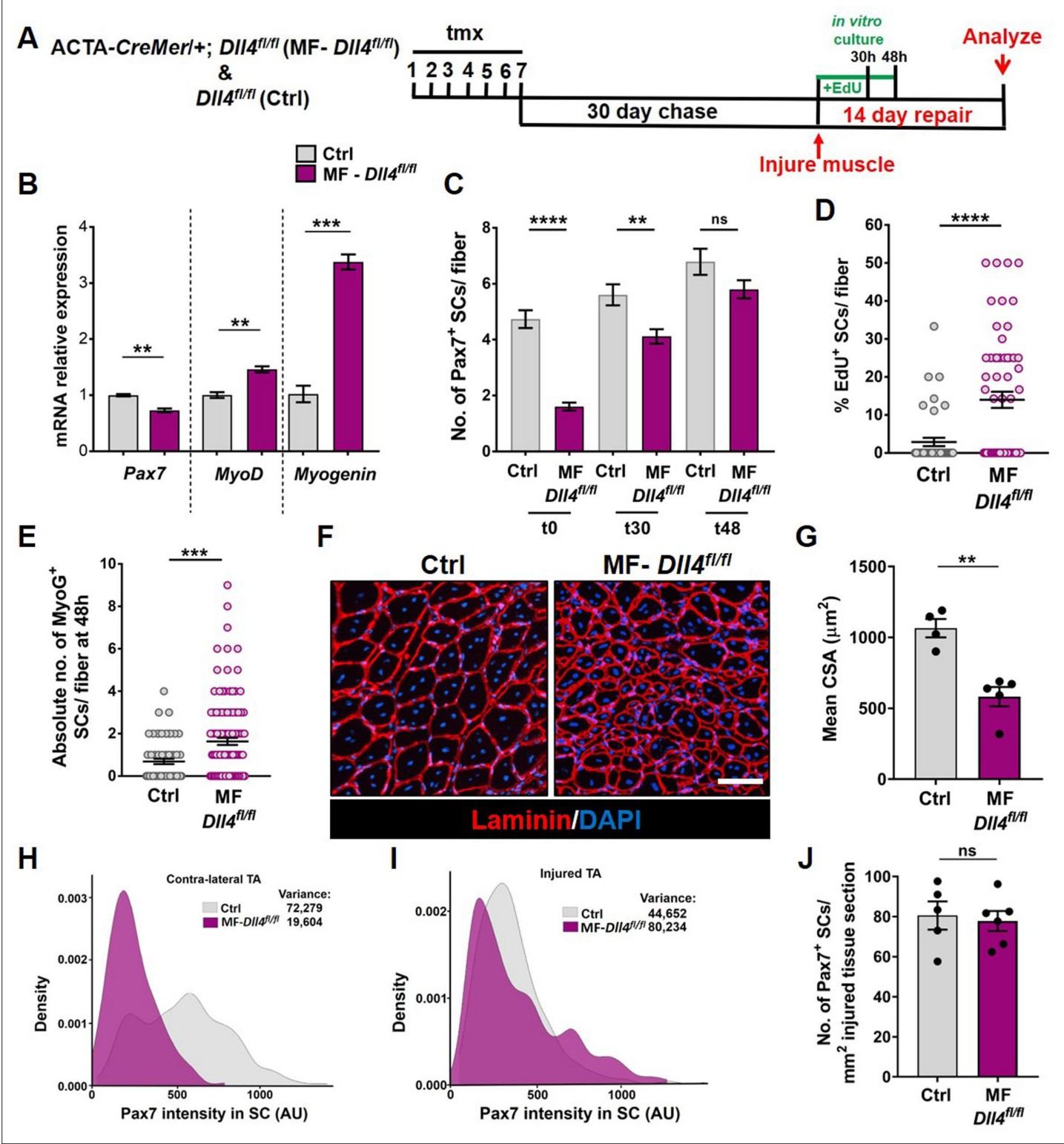

**Figure 5.** Niche-derived Dll4 constrains the proliferative and commitment potential of the quiescent muscle stem cell (QSC) pool. (**A**) Schematic representation of the experimental design. (**B**) qRT-PCR for *Pax7*, *MyoD*, and *Myogenin* transcripts in freshly isolated stem cells (SCs) from Control and MF-*Dll4*<sup>fl/fl</sup> fibers (*n* = 2). (**C**) Number of Pax7+ SCs at t0, t30, and t48 in cultured Control and MF-*Dll4*<sup>fl/fl</sup> fibers (*n* = 3). (**D**) Control and Dll4 deleted fibers were cultured in plating media containing EdU for 30 hr and the percent of EdU+ SCs per fiber were quantified (*n* = 3). (**E**) Control and MF-*Dll4*<sup>fl/fl</sup> fibers were cultured in vitro for 48 hr and the MyoG+ cells per fiber were quantified (*n* = 3). (**F, G**) Representative images (in F) and quantification (in G) of mean cross-sectional area of Control and MF-*Dll4*<sup>fl/fl</sup> tibialis anterior (TA) fibers, injured and regenerated for 14 days (*n* ≥ 4). (**H, I**) Density map of Pax7 intensity

*Figure 5 continued on next page*

*Figure 5 continued*

in SCs from Ctrl and MF-*Dll4^{fl/fl}* contralateral TA (in H) and injured TA (in I), 14-day postinjury (*n* = 3). (**J**) Number of Pax7^+ SCs in regenerated TA muscle, 14 days after injury (*n* ≥ 5). Error bars, mean ± standard error of the mean (SEM); ns, nonsignificant, **p < 0.01, ***p < 0.001, ****p < 0.0001; scale bars, 100 μm in (**F**).

The online version of this article includes the following source data and figure supplement(s) for figure 5:

**Source data 1.** Niche derived Dll4 constrains the proliferative and commitment potential of the QSC pool.

**Source data 2.** Niche derived Dll4 constrains the proliferative and commitment potential of the QSC pool.

**Source data 3.** Niche derived Dll4 constrains the proliferative and commitment potential of the QSC pool.

**Source data 4.** Niche derived Dll4 constrains the proliferative and commitment potential of the QSC pool.

**Source data 5.** Niche derived Dll4 constrains the proliferative and commitment potential of the QSC pool.

**Source data 6.** Niche derived Dll4 constrains the proliferative and commitment potential of the QSC pool.

**Source data 7.** Niche derived Dll4 constrains the proliferative and commitment potential of the QSC pool.

**Source data 8.** Niche derived Dll4 constrains the proliferative and commitment potential of the QSC pool.

**Source data 9.** Niche derived Dll4 constrains the proliferative and commitment potential of the QSC pool.

**Figure supplement 1.** Niche depletion of *Dll4* causes stem cells (SCs) to proliferate faster in the presence of mitogen.

**Figure supplement 2.** Impaired regenerative potential of stem cells (SCs) on Dll4-depleted niche.

niche (*Figure 6C, F, G*). RT-qPCR of myogenic markers shows that *Pax7* and *Myod* transcript levels did not change. However, *Myog* was upregulated in the SCs remaining on a Mib1-depleted niche (*Figure 7D*). In response to activation cues, SCs on a Mib1-depleted niche proliferate faster and upregulate Myogenin compared to the SCs on control muscle fibers, indicating that the remaining SCs are primed for proliferation and differentiation (*Figure 7E–G*). In response to muscle injury, the size of regenerating muscle fibers was smaller than the control muscle fibers, although the number of Pax7^+ SCs is the same as control in the regenerated muscle (*Figure 7H–J*). Therefore, the effect of Mib1 deletion from the niche mimics that of Dll4 deletion. In conclusion, niche-derived Mib1 expressed in a heterogeneous pattern along the muscle fiber maintains the normal continuum of diverse QSC states during tissue homeostasis.

## Discussion

The composition and location of niche cells are critical for SC regulation. The multinucleated muscle fiber acts as a niche cell to maintain the QSC pool in a continuum of diverse states through the heterogeneous patterning of the E3 ligase, Mib1 and activation of Dll4, in a fiber-autonomous manner. Therefore, niche-derived Dll4 acts as a rheostat, placing a cell along a continuum of states that are fate biased (*Figure 8*).

Our results provide the first direct demonstration of a Notch ligand from a specific cell type that is critical to maintain a continuum of metastable states within the QSC pool. This diversity and plasticity are important in the event of an injury, so that all the SCs are not equally lost to activation or differentiation. This continuum is actively maintained in uninjured muscle by the heterogeneous pattern (and levels) of Dll4 and Mib1 on the muscle fibers. Depletion of Dll4 and Mib1 in muscle fibers causes a contraction of the SC continuum toward more homogeneous committed states.

The spatial localization of the different factors in the cytoplasm of a single multinucleated cell governs the cell state of the SC pool. Expression of Wnt4 along the fiber is ubiquitous, thereby inhibiting proliferation of all SCs (*Eliazer et al., 2019*). The heterogeneous expression of Dll4 within the muscle fiber allows the maintenance of a continuum of differentiated states, with SCs in a region of low Dll4 expression are primed to proliferate and differentiate when subjected to injury in vivo or exposed to mitogen in vitro.

Although snRNA sequencing shows transcriptional heterogeneity within the myonuclei of single muscle fibers (*Kim et al., 2020*; *Petrany et al., 2020*), we show that the transcripts of Dll4 are uniformly expressed along the muscle fiber and the protein expression is heterogeneous. This highlights the importance of protein heterogeneity within single muscle fibers. The varied expression of Dll4 is established by the E3 ligase, Mib1 that adds ubiquitin groups to Dll4 and activates them by a

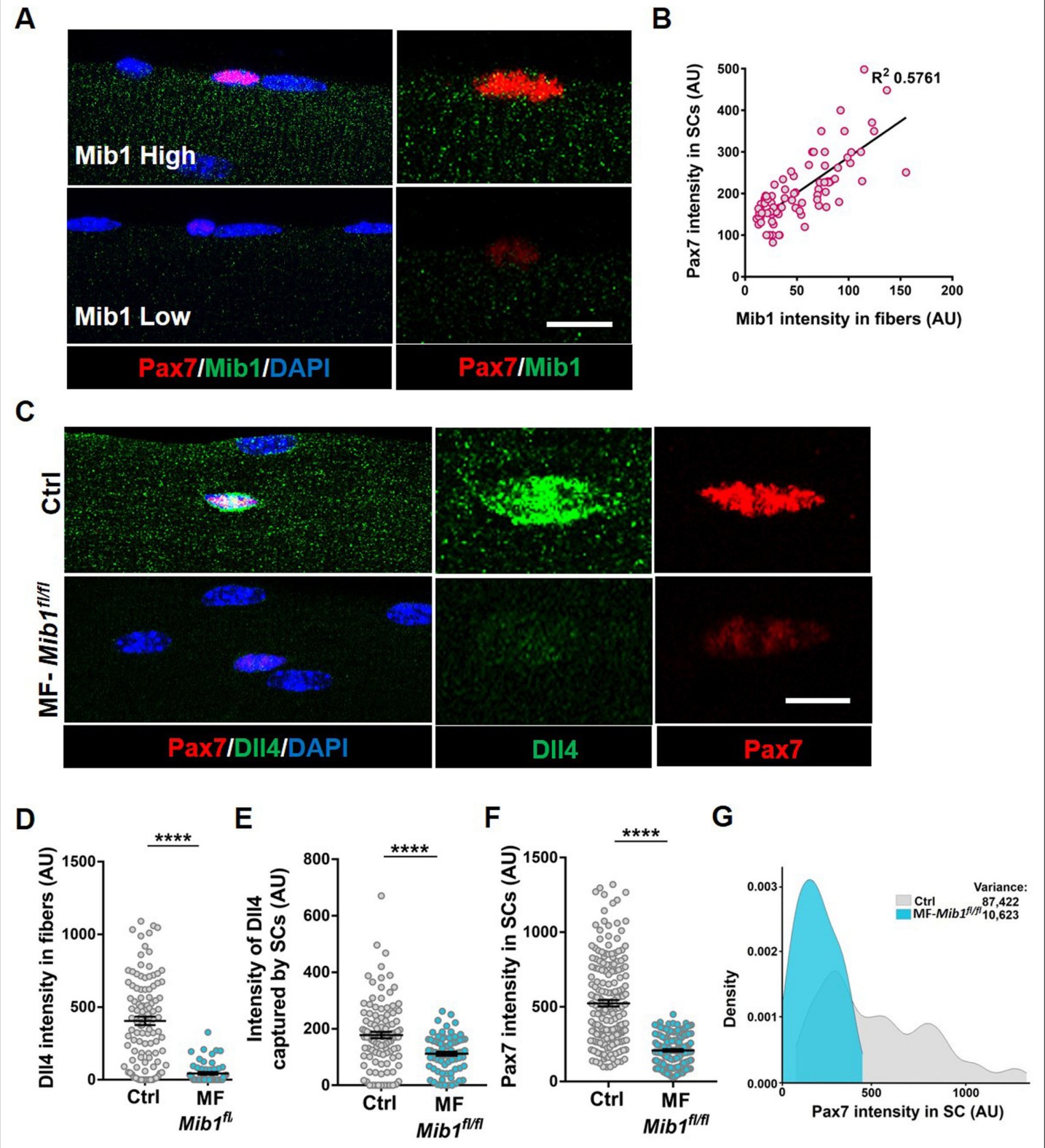

**Figure 6.** Niche-derived Mib1 directs spatial patterning and activation of Dll4. (**A**) Representative images of WT fibers showing regions of high and low Mib1 protein expression and the adjacent Pax7⁺ stem cell (SC). (**B**) A bivariate plot showing Mib1 intensity in the fiber and Pax7 intensity in the adjacent SC. (**C–G**) Representative images of Dll4 expression in fibers and Pax7 expression in adjacent SCs present on Control and MF-*Mib1^{fl/fl}* fibers (in C), quantification of Dll4 intensity in fibers (in D), intensity of Dll4 captured by SCs (in E), Pax7 intensity in SCs (in F), and a density map of Pax7 intensity in

*Figure 6 continued on next page*

Figure 6 continued

SCs (in G) on Control and MF-*Mib1^fl/fl* fibers (*n* = 3 mice). Error bars, mean ± standard error of the mean (SEM); ****$p < 0.0001$; scale bars, 5 μm in (**A**) and (**C**).

The online version of this article includes the following source data for figure 6:

**Source data 1.** Niche-derived Mib1 directs spatial patterning and activation of Dll4.

**Source data 2.** Niche-derived Mib1 directs spatial patterning and activation of Dll4.

**Source data 3.** Niche-derived Mib1 directs spatial patterning and activation of Dll4.

process of endocytosis (*Koo et al., 2005*; *Koo, 2007*). The formation of endosomes might play a role in clustering the Dll4 ligand to present them to the adjacent receptors on the QSCs.

Previous work has shown that Mib1 is essential in postnatal developing muscle to convert proliferating SCs into a quiescent state. Sex hormones activate Notch signaling pathway during puberty, thereby driving SCs into quiescence (*Kim et al., 2016*). Here, we show a different function of Mib1 in adult muscle under homeostatic conditions. In adult muscle fibers, Mib1 regulates the spatial expression and levels of Dll4, and is required to maintain a gradient of commitment states within the QSC pool.

The deletion of *Mib1* or *Dll4* within the fiber resulted in a 50% diminution of the SC pool, whereas, deletion of *Rbpj* in adult SCs displayed a more robust depletion (>95%) (*Bjornson et al., 2012*; *Mourikis et al., 2012*). This suggests that Rbpj might have targets other than the Notch signaling pathway. Deletion of *Rbpj* in SCs also caused the SC pool to precociously differentiate. We propose that SCs sit along a continuum dictated by the levels of fiber-derived Dll4. Disruption of Dll4 and Mib1 levels in the muscle fiber repositions the SCs along the continuum, biasing cell fate outcomes of the whole population.

Notch ligands added to SCs in vitro promoted quiescence of myoblasts at the expense of differentiation, but did not restore stemness (*Sakai et al., 2017*). Therefore, quiescence is not equivalent to stemness; consistent with the observations that only subsets of QSCs possess self-renewal potential in transplantation assay (*Rocheteau et al., 2012*; *Chakkalakal et al., 2012*; *Sacco et al., 2008*). These data also raise the possibility that Notch signaling directly regulates differentiation potential of a QSC rather than direct regulation of the quiescent state.

scRNA-seq on cells from regenerating muscle suggests that Dll1 is expressed in a subset of differentiating, Myogenin⁺ cells and is important for self-renewal (*Yartseva et al., 2020*). Dll1 expressed in activated and differentiated myogenic cells regulates the self-renewal of neighboring myogenic cells through a process of lateral inhibition (*Zhang et al., 2021*). In vitro experiments using co-culture of primary myoblasts with 3T3 cells overexpressing Dll4 (*Low et al., 2018*), or with endothelial cells (*Verma et al., 2018*) suggests the importance of the Notch ligand Dll4 in self-renewal at the expense of differentiation. We posit that in the presence of extrinsic cues that drive cell cycle exit (quiescence or differentiation) cells will be fated to quiescence (self-renewal) or differentiation based on location along a continuum. Notch signaling maintains cells in metastable states competent for a return to quiescence.

Using a low-dose tamoxifen strategy, we were able to demonstrate that decreasing Dll4 shifted the continuum toward commitment, without any change in SC number in the absence of injury. Therefore, QSC pool exists in a series of metastable states that is under the control of niche-derived Dll4. Prior work demonstrated that SCs with low levels of Pax7 are less competent for self-renewal in vivo (*Rocheteau et al., 2012*). Based on the decrease in Pax7 levels after Dll4 depletion from the niche, we predicted a diminution of the SC pool in response to injury. Instead, the number of SCs and Pax7 levels was increased after injury; restoring them back to control levels (*Figure 5H, I*). This demonstrates a level of plasticity and bidirectional flow of SCs along the continuum to allow the reequilibrium of metastable states after tissue repair is complete. This interpretation is only possible because we developed an approach that allows for the transient loss of Dll4 in the niche prior to injury.

In the future, it will be interesting to investigate how distinct niche factors that control quiescence versus commitment are coordinated. Our findings provide proof-of-principle for niche-based strategies to modify the fate potential of SCs while residing in a quiescent state.

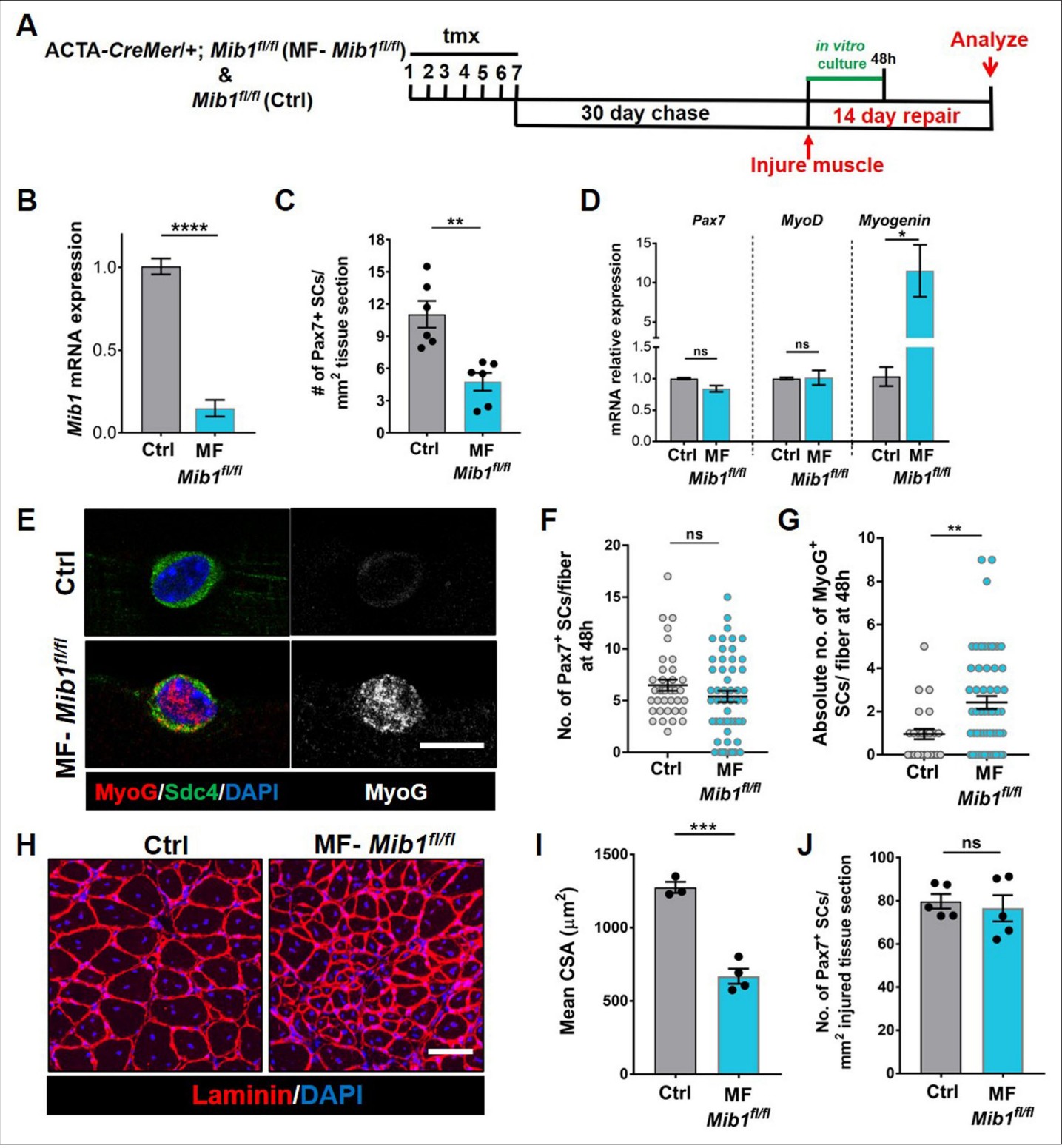

**Figure 7.** Mib1 in the muscle fiber maintains quiescent muscle stem cells (QSCs) by inhibiting differentiation. (**A**) Schematic representation of the experimental design. (**B**) qRT-PCR for *mib1* transcripts in fibers of Control and MF-*Mib1^fl/fl^* normalized to GAPDH (*n* = 3). (**C**) Number of Pax7+ stem cells (SCs) in tibialis anterior (TA) muscle sections of Control and MF-*Mib1^fl/fl^* (*n* = 6). (**D**) qRT-PCR for *Pax7*, *MyoD*, and *Myogenin* transcripts in freshly isolated SCs from Control and MF-*Mib1^fl/fl^* fibers, normalized to *GAPDH* (*n* = 3). (**E–G**) Representative images of Sdc4 and MyoG expression in SCs from Control and Mib1-deleted fibers, cultured in vitro in plating media for 48 hr (in E), the number of Pax7+ SCs (in F), and the absolute number of MyoG+ cells (in G) per fiber were quantified (*n* = 3). (**H, I**) Representative images (in H) and quantification (in I) of mean cross-sectional area of Control and MF-*Mib1^fl/fl^*

*Figure 7 continued on next page*

*Figure 7 continued*

TA muscle fibers, injured and regenerated for 14 days ($n \geq 3$). (**J**) Number of Pax7$^+$ SCs in regenerated TA muscle, 14 days after injury ($n = 5$). Error bars, mean ± standard error of the mean (SEM); ns, nonsignificant, *$p < 0.05$, **$p < 0.01$, ***$p < 0.001$, ****$p < 0.0001$; scale bars, 10 µm in (**E**) and 100 µm in (**H**).

The online version of this article includes the following source data for figure 7:

**Source data 1.** Niche-derived Mib1 directs spatial patterning and activation of Dll4.

**Source data 2.** Niche-derived Mib1 directs spatial patterning and activation of Dll4.

**Source data 3.** Niche-derived Mib1 directs spatial patterning and activation of Dll4.

**Source data 4.** Niche-derived Mib1 directs spatial patterning and activation of Dll4.

**Source data 5.** Niche-derived Mib1 directs spatial patterning and activation of Dll4.

**Source data 6.** Niche-derived Mib1 directs spatial patterning and activation of Dll4.

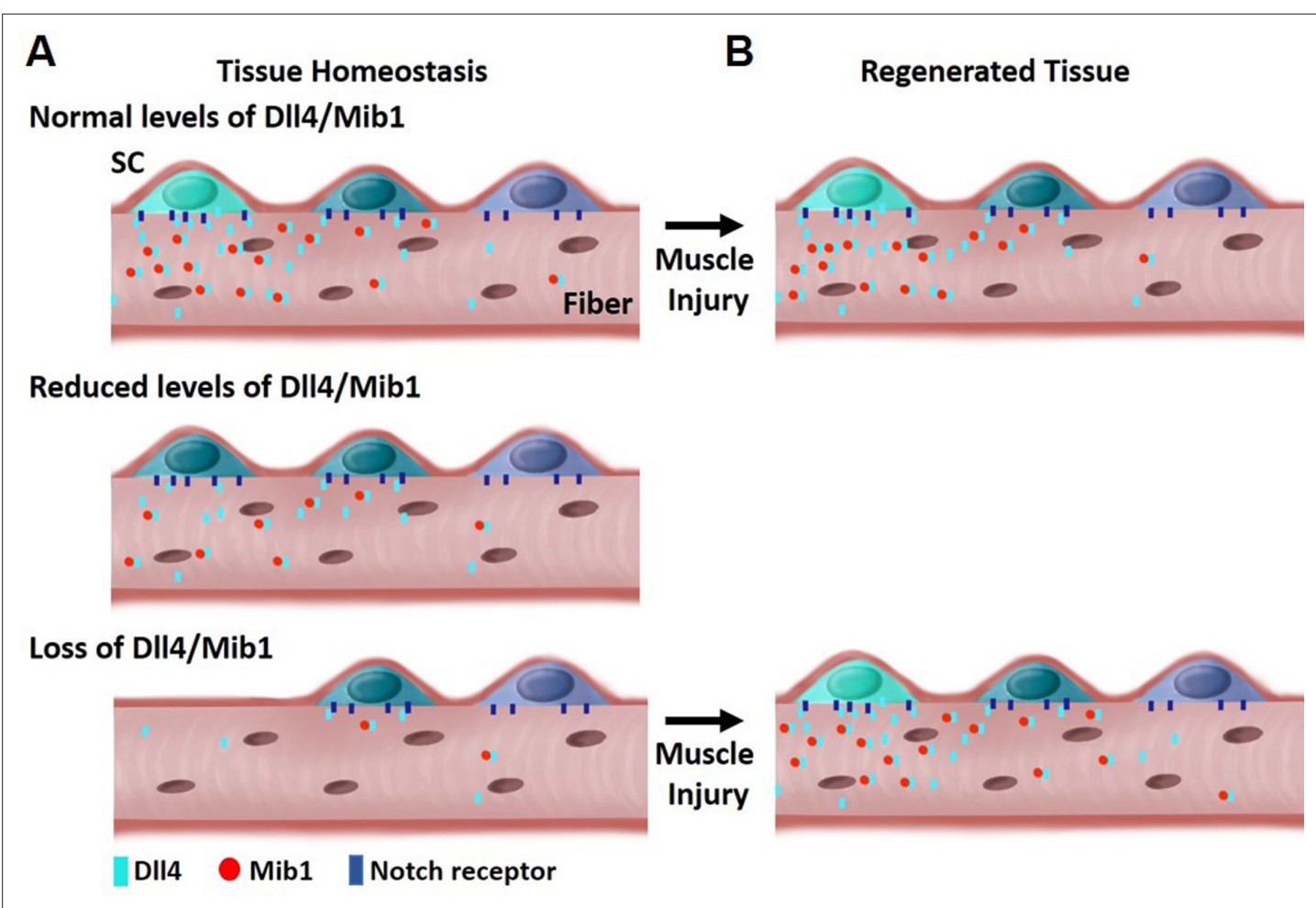

**Figure 8.** Model depicting heterogeneous levels of Dll4/Mib1 in the niche regulates a continuum of metastable stem cell (SC) states. (**A**) Top: During tissue homeostasis, Notch ligand Dll4 is present in a heterogeneous manner along the length of the muscle fibers that maintains the size of the SC pool and a continuum of quiescent metastable cell states. Middle: A reduction in Dll4/Mib1 levels does not change the number of SCs, but causes a shift in metastable states toward activation and commitment. Bottom: Complete ablation of Dll4/Mib1 in the muscle fibers results in a more dramatic shift in the metastable states away from quiescent self-renewing fates toward more proliferative and committed fates, and SC loss likely by fusion into muscle fibers. (**B**) Top: In the event of an injury, the SC metastable states are reformed on newly regenerated muscle fibers. Bottom: After injury to a Dll4/Mib1-depleted niche, wildtype SCs reform the niche, restoring the normal distribution of metastable SC states.

# Materials and methods

## Animals

Mice were housed and maintained in accordance with the guidelines of the Laboratory Animal Research Center (LARC) of University of California, San Francisco. C57BL/6 were obtained from Jackson Laboratory. Previously published *Pax7-nGFP* (**Rocheteau et al., 2012**), human *ACTA1-CreMer* (**McCarthy et al., 2012**), *Dll4$^{flox/flox}$* (**Hozumi et al., 2008**), *Mib1$^{flox/flox}$* (**Koo, 2007**), *Pax7-CreER* (**Nishijo et al., 2009**), and Notch reporter *CBF1-GFP* (**Duncan et al., 2005**) were used in this study. All mice used for experiments were adults, between 12 and 16 weeks of age. The control and experimental mice used are littermates in all experiments. Approximately equal numbers of male and female mice were used in all experiments. Animals were genotyped by PCR using tail DNA. Primer sequences are available upon request.

## Animal procedures

Tamoxifen (tmx, Sigma) was dissolved in corn oil at a concentration of 20 mg/ml. Both control and experimental mice were administered tamoxifen at a concentration of 150 mg/kg/day for seven continuous days by intraperitoneal injection. The mice were left to chase for 30 days before analysis. To reduce (not ablate completely) the levels of Dll4 in the muscle fibers, tmx was IP injected at a concentration of 75 mg/kg/day for seven consecutive days.

## Muscle injury

Control and experimental mice were anesthetized by isofluorane inhalation and 50 μl of 1.2% BaCl$_2$ was injected into and along the length of the tibialis anterior (TA) muscle. After 14 and 40 days of regeneration, mice were euthanized, the contralateral uninjured TA and injured TA muscle were fixed immediately in 4% PFA(Paraformaldehyde) and frozen in 20% sucrose/OCT medium. 8-μm cross-sections of the muscle were made and stained for anti-laminin. ×10 images were collected at three regions in the mid-belly of each muscle. Only mice that had more than 80% injury in their TA were analyzed. All the regenerating fibers in the entire TA section were analyzed for fiber size. The average cross-sectional area of the fibers was determined using ImageJ software.

## Isolation of single muscle fibers

Single muscle fibers were isolated from the EDL muscle of the adult mouse as described previously (**Eliazer et al., 2019**). The single fibers were fixed immediately in 4% PFA for 10 min or cultured in plating media (DMEM(Delbecco's Modified Eagle Medium) with 10% horse serum).

For in vitro cell cycle entry assays, single muscle fibers from control and Myofiber-*Dll4$^{fl/fl}$* mice were harvested and cultured in plating media containing EdU (10 μm; Carbosynth) for 30 hr. For in vitro differentiation assay, the single muscle fibers from control, Myofiber-*Dll4$^{fl/fl}$* and Myofiber-*Mib1$^{fl/fl}$* were cultured in plating media for 48 hr. The fibers were then fixed and stained for different antibodies. EdU staining was done using the Click-iT Plus EdU Alexa Fluor 594 Imaging Kit (Invitrogen) followed by staining with anti-Pax7 antibody.

## Isolation of SCs and fluorescence-assisted cell sorting

Satellite cells were isolated from hindlimb and forelimb muscles as previously described (**Eliazer et al., 2019**). The mononuclear muscle cells were stained for PE-Cy7 anti-mouse CD31 (clone 390; BD Biosciences), PE-Cy7 anti-mouse CD45 (clone 30-F11; BD Biosciences), APC-Cy7 anti-mouse Sca1 (clone D7; BD Biosciences), PE anti-mouse CD106/VCAM-1 (Invitrogen), and APC anti-α7 integrin (clone R2F2; AbLab). Fluorescence-assisted cell sorting (FACS) was performed using FACS Aria II (BD Biosciences) by gating for CD31$^-$/CD45$^-$/Sca1$^-$/α7 integrin$^+$/VCAM1$^+$ to isolate SCs. SCs from Pax7-nGFP mouse were sorted for GFP fluorescence. The GFP$^+$ gate was divided into top 15% (GFP$^{high}$ fraction), middle 45% (Pax7$^{medium}$ fraction), and bottom 15% (Pax7$^{low}$ fraction). The isolated SCs were fixed immediately at t0 or cultured in growth media (Ham's F10 media, 20% fetal bovine serum, 5 ng/ml FGF2) containing 10 μm EdU for 60 hr (cell cycle entry assay) or cultured in plating media (DMEM with 10% horse serum) for 3 days (differentiation assay).

## Immunostaining

Fixed myofibers were permeabilized with 0.2% Triton X-100/phosphate-buffered saline (PBS) and blocked with 10% goat serum/0.2% Triton X/PBS. Primary antibodies used in this study were: mouse

anti-Pax7 (DSHB), rabbit anti-Laminin (Abcam), rabbit anti-Dll4 (Thermo Fisher Scientific), rabbit anti-Mib1 (Sigma), rabbit anti-Syndecan4 (Abcam), rabbit anti-DDX6 (Bethyl Laboratories Inc), rabbit anti-Myogenin (Santa Cruz Biotechnology), and DAPI(4',6-diamidino-2-phenylindole) (Life Technologies). Primary antibodies were visualized with fluorochrome conjugated secondary antibodies (Invitrogen). The stained fibers were mounted in Fluoromount-G mounting medium (SouthernBiotech). For most of the staining, the images were taken using a ×20 Plan Fluor objective of the Nikon Eclipse Ti microscope. Anti-Pax7 and anti-Ddx6 stained images were obtained with a ×40 Plan Fluor objective of the same microscope. Dll4 and Mib1 stained images were taken with ×40 oil objective of Leica DMi8 Confocal Microscope. 15 µm z-stacks around the Pax7$^+$ SC were taken and the sum projection of the images was obtained. The filter settings, gain and exposure values were kept constant between experiments. The intensity of expression is determined by manually drawing a region of Interest (ROI) on a Pax7-positive SC. This will give the mean pixel intensity of the ROI in all the channels. The ROI is copied onto another region where there is no Pax7-positive cell to calculate the background intensity. The background intensity is subtracted and the mean intensity is plotted with GraphPad Prism 7. To determine the intensity of Dll4 captured by SC, a ROI is drawn around the Pax7$^+$ SC. To determine the intensity of Dll4 in the muscle fibers adjacent to the SCs, a ROI is drawn in an area of 100 µm$^2$ surrounding the SC. Representative images for antibody staining were taken using Leica DMi8 Confocal Microscope.

## RNAscope

RNAscope for *Dll4* was performed on fixed myofibers as previously described (*Kann and Krauss, 2019*). Briefly the single muscle fibers are isolated, fixed with 4% PFA and dehydrated with 100% methanol. The fibers are then rehydrated with a series of decreasing concentrations of methanol and PBS with 0.1% Tween 20 followed by protease digestion, hybridization of the RNA probe, amplification of signal and conjugation with fluorophore.

## Quantitative PCR

Total RNA was isolated from single muscle fibers (from EDL muscle), or SCs using Trizol (Invitrogen) according to the manufacturer's protocol. The RNA was DNAse treated using Turbo DNA free kit (Life Technologies). cDNA was synthesized from RNA using the Superscript First Strand Synthesis System (Invitrogen). Quantitative PCR (qPCR) was performed in triplicates from 5 ng of RNA per reaction using Platinum SYBR Green qPCR Super Mix-UDG w/ROX (Invitrogen) on a ViiA7 qPCR detection system (Life Technologies). All reactions for RT-qPCR were performed using the following conditions: 50°C for 2 min, 95°C for 2 min, 40 cycles of a two-step reaction of denaturation at 95°C for 15 min, and annealing at 60°C for 30 s. The mean Ct values from triplicates were used in the comparative 2$^{-\Delta\Delta Ct}$ method. To analyze the expression of Notch ligands on the adult muscle fiber, 2$^{-\Delta Ct}$ method was used. The results were normalized to GAPDH mRNA controls. The primers used in this study are listed in *Supplementary file 2*.

## Microarray

Total RNA was isolated from single muscle fibers of the EDL muscle from postnatal day 3 (p3), postnatal day 7 (p7), and adult mouse hindlimb using TRIzol reagent (Invitrogen). The processing of RNA, hybridization, and analysis is previously described (*Eliazer et al., 2019*). The expression values for Notch ligands are log2 transformed and listed in *Supplementary file 1*.

## Quantification and statistical analysis

The density maps for Pax7 and Ddx6 intensity were drawn using R. The variance of Pax7 and Ddx6 intensity levels across the SC pool is calculated as the square of standard deviation. The statistical details of experiments can be found in the figure legends. No statistical methods were used to predetermine sample size. The investigators were not blinded to allocation during experiments and outcome assessment. No animal was excluded from analysis. All data are represented as mean ± standard error of the mean. Significance was calculated using the two-tailed unpaired Student's *t*-tests (GraphPad Prism 7). The number of replicates (*n*) for each experiment is indicated in the figure legends. Differences were considered statistically different at p < 0.05.

## Acknowledgements

We would like to thank Drs. Karyn Esser, Charles Keller, Young-Yun Kong, and Sonoko Habu for providing transgenic mice. We would like to thank Hallie Nelson for technical assistance, and members of the Brack laboratory for critical discussions during the preparation of this manuscript. We acknowledge the UCSF Parnassus Flow Cytometry Core (RRID:SCR_018206) supported in part by Grant NIH P30 DK063720 and by the NIH S10 Instrumentation Grant S10 1S10OD021822-01. This work was supported by NIH grants (R01AR060868, R01AR061002, R01AR076252) to ASB, and (F32AR067594) to SE.

## Additional information

### Competing interests

Andrew S Brack: Reviewing editor, *eLife*. The other authors declare that no competing interests exist.

### Funding

| Funder | Grant reference number | Author |
| --- | --- | --- |
| National Institutes of Health | R01AR060868 | Andrew S Brack |
| National Institutes of Health | R01AR061002 | Andrew S Brack |
| National Institutes of Health | R01AR076252 | Andrew S Brack |
| National Institutes of Health | F32AR067594 | Susan Eliazer |
| National Institutes of Health | P30 DK063720 | Andrew S Brack |
| NIH | S10 1S10OD021822-01 | Andrew S Brack |

The funders had no role in study design, data collection, and interpretation, or the decision to submit the work for publication.

### Author contributions

Susan Eliazer, Formal analysis, Investigation, Writing - original draft, Writing - review and editing; Xuefeng Sun, Formal analysis, Methodology; Emilie Barruet, Formal analysis; Andrew S Brack, Funding acquisition, Writing - review and editing

### Author ORCIDs

Emilie Barruet http://orcid.org/0000-0002-4593-024X
Andrew S Brack http://orcid.org/0000-0002-8798-7084

### Ethics

This study was performed in strict accordance with the recommendations in the Guide for the Care and Use of Laboratory Animals of the National Institutes of Health. All of the animals were handled according to approved institutional animal care and use committee (IACUC) protocols (#AN174604, #AN176815) of the University of California San Francisco. Every effort was made to minimize suffering.

### Decision letter and Author response

Decision letter https://doi.org/10.7554/eLife.68180.sa1
Author response https://doi.org/10.7554/eLife.68180.sa2

## Additional files

### Supplementary files

• Supplementary file 1. Microarray expression of Notch ligands in p3, p7, and adult muscle fibers

(log2 values). This table includes the expression (as log2 values) of Notch ligands in postnatal day 3, postnatal day 7, and adult. Related to *Figure 2—figure supplement 1*.

• Supplementary file 2. Primers used for quantitative RT-PCR. This table includes the forward and reverse primer sequences of genes amplified by qRT-PCR. Related to *Figure 2—figure supplement 1*, *Figures 5 and 7*.

• Transparent reporting form

### Data availability

We have cited a data repository in our data availability statement, 'The microarray data generated during this study is available at NCBI GEO: GSE135163'. It is a microarray dataset from postnatal d3, d7, and adult purified single muscle fibers.

The following previously published dataset was used:

| Author(s) | Year | Dataset title | Dataset URL | Database and Identifier |
|---|---|---|---|---|
| Brack A, Eliazer S | 2019 | Expression data from wild-type single muscle fibers | https://www.ncbi.nlm.nih.gov/geo/query/acc.cgi?acc=GSE135163 | NCBI Gene Expression Omnibus, GSE135163 |

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
