## [Editor Report]

This is a strategic paper of relevance to both muscle and stem cell biologists. It bears on the generation of muscle stem cell diversity, and the factors bearing on this. Specifically, this paper identifies a particular, Notch ligand Dll4, as a myofiber-derived regulator of muscle stem cells.

---

## [Decision Letter]

**Decision letter after peer review:**

Thank you for submitting your article "Spatial heterogeneity of Dll4 within a multinucleated niche cell maintains muscle stem cell diversity" for consideration by *eLife*. Your article has been reviewed by 3 peer reviewers, and the evaluation has been overseen by a Reviewing Editor and Mone Zaidi as the Senior Editor. The following individuals involved in review of your submission have agreed to reveal their identity: Robert S. Krauss (Reviewer #1); Bradley B Olwin (Reviewer #3).

Eliazer et al. report that the Notch ligand Dll4 produced by myobers regulates the fate of muscle stem cells. Identification of such stem cell niche factors is important, and the results are interesting and beneficial to the field. The conclusion that heterogeneous distribution of Dll4 on myofibers maintains a continuum of muscle stem cell fates is, however, not sufficiently supported by the data and therefore premature. While some additional experimentation is asked for, this and most other comments by the reviewers can be addressed by alterations to the text.

Essential revisions:

1) Changes to the text are necessary to temper the conclusion that Dll4 signals to maintain a continuum of SC fates and raise alternative interpretations to the data. Additionally, prior work demonstrating that Pax7 protein levels are variable and that the level of Pax7 affects satellite cell fate has been published and specific early papers should be cited (Zammit, JCB 2004; and Olguin, Dev Bio 2004). The title should also be reworded to reflect these changes. Please see the individual reviews for more specific comments on how to address this issue.

2) *Mib1* regulates multiple Notch ligands. Could *Mib1* deletion affect Notch ligands other than Dll4 that are also expressed by myofibers? This can be addressed by IF of single myofibers for additional Notch ligands reported in Figure S2B.

3) Expression of Notch pathway target genes should be examined in satellite cells from control and MF-Dll4 mice by qRT-PCR. This will presumably validate Dll4's expected role in maintaining quiescence-promoting Notch signaling and help address the observation that the MF-Dll4 phenotype is less pronounced than the satellite cell-specific RBP-J knockout phenotype.

4) Additional changes to the text are requested in the comments from individual reviewers.

*Reviewer #1 (Recommendations for the authors):*

1. Given the known role of Dll4 as a Notch ligand, it is highly likely that the non-autonomous SC phenotypes are a consequence of reduced Notch pathway activity in these cells. It is important to show that reduced Notch pathway activity has actually occurred. I suggest that the authors FACS sort SCs from control and MF-Dll4 mice and measure the levels of Notch target genes by qRT-PCR. This technique was used in the paper, and a number of validated Notch target genes in SCs have been published.

2. The conclusions that the amount of Dll4 surrounding individual SCs correlates with Pax7 levels, and that the distribution of Pax7 levels is left-shifted in the absence of Dll4 in fibers, are convincing. High Pax7 levels, as measured with a Pax7-nGFP transgene, have been reported to correlate with greater SC dormancy (Rocheteau et al.). However, the various cells measured in this paper were tested in vitro and not in vivo for true stem cell activity, including self-renewal. Therefore, the conclusion about Dll4 controlling "a continuum of quiescent cell states ranging from deep quiescent, non-committed states to more committed states" (lines 238-239 in the Discussion) seems too strong. Simply tempering the conclusion would suffice.

3. Figure S1 and Materials and methods: Figure S1D shows that Pax7-nGFP(High) cells incorporate EdU less efficiently than the (Medium) or (Low) cells. A very small percentage of cells overall incorporated EdU. The legend say the cells were in plating medium (10% horse serum) but the Methods section says they were in growth medium (20% FBS plus FGF2). Which is correct? The low percentage of EdU+ cells is more consistent with the former. If so, have the authors tried the latter?

4. Figure 4 and lines 155-163: Numbers of SCs on single fibers from control and MF-Dll4 mice are similar after culture for 48 hr in 10% serum-containing medium (Figure 4E). Based on the Methods section, I assume this is 10% horse serum (which is mitogen poor), rather than growth medium. The authors conclude that there was compensatory cell proliferation of remaining MF-Dll4 SCs, as there are fewer of these at 30 hr. However, the SC numbers at 48 hr average 5-6 per EDL fiber, which is a typical number found associated with freshly isolated EDL fibers. After 30 hr, the numbers are similar (there are slightly fewer SCs on MF-Dll4 fibers but the difference is small – about one SC per fiber less, on average). It seems to me that there may be very little cell proliferation occurring at all on these fibers, consistent with use of horse serum (if that is indeed the case). It would also be consistent with the published in vivo phenotype of mice lacking RBP-J in SCs. RBP-J-null SCs often incorporate EdU but withdraw from the cell cycle prior to dividing, express myogenin, and precociously differentiate. This phenotype seems more consistent with the data on MF-Dll4 SCs in Figures 4B-F, and would further link the similarity between loss of fiber Dll4 and loss of Notch signaling in SCs.

5. Figure 5: The area of high *Mib1* protein seems wide relative to the intensity of Dll4 around SCs but still correlates with Pax7 levels in "adjacent" SCs. What area defines "adjacent"?

6. The textbook view of *Mib1* is that it monoubiquitylates Notch ligands in complex with Notch, leading to ligand endocytosis and Notch receptor activation. In this case, it also seems to play a major role in overall Dll4 localization. It would be worth citing another example of this, as it is not always the case (see for example, Cell Reports 19, 351-363, 2017).

7. Lines 264-265: Because the MF-Dll4 phenotype is less pronounced than the SC RBP-J knockout phenotype, the authors suggest that RBP-J may have targets other than the Notch pathway. Doesn't it seem more likely that Dll4 is not the only relevant Notch ligand produced by the fiber? Figure S2B shows similar levels of mRNA for additional Notch ligands ae expressed by fibers. The experiment requested in comment 1 will help here too. If reduction of Notch target gene expression is incomplete in MF-Dll4 SCs (relative to that reported with RBP-J removal from SCs), it would be consistent with additional ligands playing a role. If reduction of Notch target gene expression was similar between the two, it would be more consistent with an additional role for RBP-J.

*Reviewer #2 (Recommendations for the authors):*

1. A conclusive demonstration demonstration that spatial distribution of Dll4 and *Mib1* in myofibers regulate diversity of MuSC state is lacking. While the authors show that such diverse spatial distribution of the two of factors exist in myofibers, that data that they are responsible for determining MuSC heterogeneity are correlative. While genetic deletion of Dll4 or *Mib1* in myofibers resulted in a loss of MuSC diversity and premature activation, this approach complete deletes expression of the proteins, it does not modulate the diversity of Dll4 protein distribution on the myofiber. The resulting effect is that MuSC are prematurely activated due to loss of Dll4-mediated Notch signaling, but it is unclear if this is due to loss of spatial distribution of the proteins as suggested by the authors.

2. Mib regulates multiple Notch ligands. Does *Mib1* deletion affects other Notch ligands, beyond Dll4, in myofibers?

3. The authors state "Dll4 expression was also variable in regions devoid of SCs, suggesting the presence of a SC does not dictate Dll4 levels along muscle fibers". What is the frequency of Dll4 diversity independent of MuSC? Is it a rare event or is as frequent as the one correlated to the presence of MuSC on the myofiber?

4. While the authors found no correlation with the distance from the NMJ, this does not necessarily mean that there is no correlation with anatomically defined regions of the myofiber. in vivo, different regions of the myofiber are exposed to different local microenvironments, thus it cannot be rule out the possibility that such a correlation exists with other anatomical locations, that are lost upon myofiber isolation from the tissue.

5. Distinction between Dll4 on myofiber and Dll4 captured by MuSC: it would be useful if the authors would include in the methods how they distinguish Dll4 on myofiber from the Dll4 captured by MuSC.

6. In Figures 3I, 4K and S5J, it would be useful to include representative image of tissue sections, to support the quantification shown.

*Reviewer #3 (Recommendations for the authors):*

If the authors widen their potential interpretations and do not argue that Dll4 regulates the relative levels of Pax7, I would be open to publishing without additional experiments. The primary observations are highly interesting and I feel the data are over interpreted.

---

## [Author Response]

Eliazer et al. report that the Notch ligand Dll4 produced by myobers regulates the fate of muscle stem cells. Identification of such stem cell niche factors is important, and the results are interesting and beneficial to the field. The conclusion that heterogeneous distribution of Dll4 on myofibers maintains a continuum of muscle stem cell fates is, however, not sufficiently supported by the data and therefore premature. While some additional experimentation is asked for, this and most other comments by the reviewers can be addressed by alterations to the text.

We would like to thank the reviewers for recognizing that identification of Dll4 as a stem cell niche factor is important to the field. We show in this manuscript that the heterogeneous distribution of Dll4 on the muscle fibers maintain stem cell diversity as a continuum of stem cell states (based on Pax7 and Ddx6 levels), that are biased to differential cell fates. What has been shown so far in the literature is the presence of Pax7^high^ and Pax7^low^ states. We show in this manuscript that there is a range of cell states from high to low based on two different stem cell markers Pax7 and Ddx6 and these states are maintained by the levels of Dll4 in the muscle fiber.

We have now performed an additional experiment where we manipulate the levels of Dll4 on the fibers by reducing the amount of tamoxifen that is given to the mice. Reducing the levels of Dll4 on the fibers, not completely deleting it, causes a leftward shift (or reduction) in the continuum of quiescent stem cell states (Figure 4-) without a diminution of stem cells.

Essential revisions:1) Changes to the text are necessary to temper the conclusion that Dll4 signals to maintain a continuum of SC fates and raise alternative interpretations to the data.

We have tempered some of the conclusions and discussed alternative interpretations. The changes are outlined below in the comments to individual reviewers.

Additionally, prior work demonstrating that Pax7 protein levels are variable and that the level of Pax7 affects satellite cell fate has been published and specific early papers should be cited (Zammit, JCB 2004; and Olguin, Dev Bio 2004).

The omission of these two seminal papers was a massive oversight on our behalf. They have now been included.

The title should also be reworded to reflect these changes.

We have modified the title to remove the spatial component of heterogeneity and to refocus on heterogeneous levels of Dll4. The original title doesn’t mention fate, rather stem cell diversity- we believe this accurately reflects the data.

The current title is: Heterogeneous levels of Δ-like 4 Within a Multinucleated Niche Cell Maintain Muscle Stem Cell Diversity.

Please see the individual reviews for more specific comments on how to address this issue.2) Mib1 regulates multiple Notch ligands. Could Mib1 deletion affect Notch ligands other than Dll4 that are also expressed by myofibers? This can be addressed by IF of single myofibers for additional Notch ligands reported in Figure S2B.

It is widely accepted that *Mib1* genetic deletion abrogates the expression of all Notch ligands at the level of post-translational regulation (Koo, B. K., et al., 2005; Koo, B. K., et al., 2007). Therefore, we suggest that *Mib1* is regulating the Notch ligands that are expressed in muscle fibers. Unfortunately, the antibodies for Notch ligands other than DLL4 are either not available or not validated. Hence, it is not possible to analyze Notch ligand expression after *Mib1* deletion.

3) Expression of Notch pathway target genes should be examined in satellite cells from control and MF-Dll4 mice by qRT-PCR. This will presumably validate Dll4's expected role in maintaining quiescence-promoting Notch signaling and help address the observation that the MF-Dll4 phenotype is less pronounced than the satellite cell-specific RBP-J knockout phenotype.

To address this question, we crossed a transgenic mouse line harboring a Notch reporter with MF-Dll4 mice to analyze Notch signaling in SCs on isolated single muscle fibers. The first experiment we performed with this reporter was to correlate the levels of Pax7 and Notch signaling on a cell-by-cell basis. In control mice, we found a linear positive relationship between the levels of Pax7 and the Notch reporter (Figure 1E, 1F). Consistent with previous reports, Notch signaling is heterogeneous.

Next, we compared Notch reporter levels in control versus Dll4-null mice. Notch reporter levels decreased to below detectable levels in Dll4 null muscle (Figure 3I). Therefore, Dll4 acts non-autonomously to regulate Notch signaling in SCs during homeostasis. Moreover, Dll4 is likely the dominant source of Notch ligands. Therefore, the residual levels of Notch signaling does not explain the milder phenotypes in the two niche depletion models (Dll4 and *Mib1*) compared to SC-specific Rbpj deletion. These data are consistent with the idea that Rbpj might have targets other than the Notch signaling pathway. We mention this in the discussion.

4) Additional changes to the text are requested in the comments from individual reviewers.

We have addressed the comments below.

Reviewer #1 (Recommendations for the authors):1. Given the known role of Dll4 as a Notch ligand, it is highly likely that the non-autonomous SC phenotypes are a consequence of reduced Notch pathway activity in these cells. It is important to show that reduced Notch pathway activity has actually occurred. I suggest that the authors FACS sort SCs from control and MF-Dll4 mice and measure the levels of Notch target genes by qRT-PCR. This technique was used in the paper, and a number of validated Notch target genes in SCs have been published.

To address this question, we crossed a transgenic mouse line harboring a Notch reporter with MF-Dll4 mice to analyze Notch signaling in FACs sorted SCs. The first experiment we performed with this reporter was to correlate the levels of Pax7 and Notch signaling on a cell-by-cell basis. In control mice, we found a linear positive relationship between the levels of Pax7 and the Notch reporter (Figure 1E, 1F). Consistent with previous reports, Notch signaling is heterogeneous.

Next, we compared Notch reporter levels in control versus Dll4-null. We observed that Notch reporter decreased to below detectable levels in Dll4 null muscle (Figure 3I). Therefore, Dll4 acts non-autonomously to regulate Notch signaling in SCs during homeostasis. Moreover, Dll4 from the muscle fiber is the dominant source of Notch ligands.

2. The conclusions that the amount of Dll4 surrounding individual SCs correlates with Pax7 levels, and that the distribution of Pax7 levels is left-shifted in the absence of Dll4 in fibers, are convincing. High Pax7 levels, as measured with a Pax7-nGFP transgene, have been reported to correlate with greater SC dormancy (Rocheteau et al.). However, the various cells measured in this paper were tested in vitro and not in vivo for true stem cell activity, including self-renewal. Therefore, the conclusion about Dll4 controlling "a continuum of quiescent cell states ranging from deep quiescent, non-committed states to more committed states" (lines 238-239 in the Discussion) seems too strong. Simply tempering the conclusion would suffice.

We were careful to not discuss self-renewal potential of the SC subsets because we did not perform the necessary experiments. However, we do clearly show that there is a continuum of states existing in a linear relationship between Pax7 and Ddx6 levels (Figure 1D). This is evidence of a continuum of different states. We used these two markers based on previous work showing their expression is high in quiescent states and decreased early during activation (Crist, C. G., et al., 2012; Zammit, P. S., et al., 2006). In addition, we provide complementary evidence that cell cycle activation rate (S-phase entry) and subsequent induction of Myogenin is related to Pax7 levels at time of isolation (Figure 1—figure supplement 1). These data mirror what was demonstrated by Rocheteau, P., et al., 2012. We have modified this sentence to “Our results provide the first direct demonstration of a Notch ligand from a specific cell type that is critical to maintain a continuum of states within the QSC pool”.

3. Figure S1 and Materials and methods: Figure S1D shows that Pax7-nGFP(High) cells incorporate EdU less efficiently than the (Medium) or (Low) cells. A very small percentage of cells overall incorporated EdU. The legend say the cells were in plating medium (10% horse serum) but the Methods section says they were in growth medium (20% FBS plus FGF2). Which is correct? The low percentage of EdU+ cells is more consistent with the former. If so, have the authors tried the latter?

Apologies for the discrepancy in the text. The Pax7-nGFP high, medium and low expressers were plated in growth medium (Ham’s F10 media containing 20% FBS and 5ng/ml FGF2). We have corrected the text in the figure legend. In addition, we have repeated the experiment with cells in culture for a longer time-point: 60h. This experiment shows that quiescent SCs sit on a continuum in a linear relationship between Pax7 expression and the time to reach S-phase when activated. This readout is often used as a surrogate for depth of quiescence (Tajbakhsh, Rando and Brack labs). Pax7-nGFP low expressers are faster to enter cycle, followed by the medium and then high GFP expressers. While the Pax7 high and low data was anticipated based on Rocheteau, P., et al., 2012, the Pax7 mid fraction was not examined in their work. Therefore, our work demonstrates a continuum based on Pax7 levels and other quiescent state markers. This new data replaces the previous figure (Figure 1—figure supplement 1D).

4. Figure 4 and lines 155-163: Numbers of SCs on single fibers from control and MF-Dll4 mice are similar after culture for 48 hr in 10% serum-containing medium (Figure 4E). Based on the Methods section, I assume this is 10% horse serum (which is mitogen poor), rather than growth medium. The authors conclude that there was compensatory cell proliferation of remaining MF-Dll4 SCs, as there are fewer of these at 30 hr. However, the SC numbers at 48 hr average 5-6 per EDL fiber, which is a typical number found associated with freshly isolated EDL fibers. After 30 hr, the numbers are similar (there are slightly fewer SCs on MF-Dll4 fibers but the difference is small – about one SC per fiber less, on average). It seems to me that there may be very little cell proliferation occurring at all on these fibers, consistent with use of horse serum (if that is indeed the case). It would also be consistent with the published in vivo phenotype of mice lacking RBP-J in SCs. RBP-J-null SCs often incorporate EdU but withdraw from the cell cycle prior to dividing, express myogenin, and precociously differentiate. This phenotype seems more consistent with the data on MF-Dll4 SCs in Figures 4B-F, and would further link the similarity between loss of fiber Dll4 and loss of Notch signaling in SCs.

I appreciate the argument the reviewer is trying to make, but we are hesitant to compare absolute SCs numbers in our experiments with those from other labs, because of the many confounding factors that will impact SC number per fiber, including genetic strain, isolation approach and batch of serum, just to name a few.

In our experiments, we find that the initial decrease in SC number at homeostasis (50%) catches up over 48 hours in mitogen. This is consistent with the increased fraction of EdU+/SCs during this window. We have reconfigured the data to help with the comparison (Figure 5C- 5D, Figure 5—figure supplement 1). If the results mimicked Rbpj, there would be a significant difference in SC number between control and null after 48 hours in culture. Therefore, deletion of Dll4 from the niche does not phenocopy SC deletion of Rbpj.

5. Figure 5: The area of high Mib1 protein seems wide relative to the intensity of Dll4 around SCs but still correlates with Pax7 levels in "adjacent" SCs. What area defines "adjacent"?

To measure the expression of Dll4 and *Mib1* on the muscle fiber we cover an area of 100μm^2^ around the SC. This is now stated in the methods. This region was determined empirically based on our qualitative observations. In Figure 2—figure supplement 2, we show the distribution of Dll4 across the entire fiber, we find no evidence that our standardized method biases the result.

6. The textbook view of Mib1 is that it monoubiquitylates Notch ligands in complex with Notch, leading to ligand endocytosis and Notch receptor activation. In this case, it also seems to play a major role in overall Dll4 localization. It would be worth citing another example of this, as it is not always the case (see for example, Cell Reports 19, 351-363, 2017).

We thank the reviewer for the suggestion, we have incorporated this reference in the manuscript.

7. Lines 264-265: Because the MF-Dll4 phenotype is less pronounced than the SC RBP-J knockout phenotype, the authors suggest that RBP-J may have targets other than the Notch pathway. Doesn't it seem more likely that Dll4 is not the only relevant Notch ligand produced by the fiber? Figure S2B shows similar levels of mRNA for additional Notch ligands ae expressed by fibers. The experiment requested in comment 1 will help here too. If reduction of Notch target gene expression is incomplete in MF-Dll4 SCs (relative to that reported with RBP-J removal from SCs), it would be consistent with additional ligands playing a role. If reduction of Notch target gene expression was similar between the two, it would be more consistent with an additional role for RBP-J.

As addressed in essential revisions comment #3 and in Reviewer 1 comment #1, we cannot exclude that other Notch ligands regulate SC quiescence during homeostasis. However, we now demonstrate that deletion of Dll4 in the muscle fibers leads to a loss of detectable Notch signaling (Figure 3I). Together, with the data showing that *Mib1* deletion in the fibers has a similar phenotype to Dll4, we conclude that Dll4 from the muscle fiber plays a dominant role in maintaining SCs during tissue homeostasis. The data is consistent with an additional role for Rbpj.

Reviewer #2 (Recommendations for the authors):1. A conclusive demonstration demonstration that spatial distribution of Dll4 and Mib1 in myofibers regulate diversity of MuSC state is lacking. While the authors show that such diverse spatial distribution of the two of factors exist in myofibers, that data that they are responsible for determining MuSC heterogeneity are correlative. While genetic deletion of Dll4 or Mib1 in myofibers resulted in a loss of MuSC diversity and premature activation, this approach complete deletes expression of the proteins, it does not modulate the diversity of Dll4 protein distribution on the myofiber. The resulting effect is that MuSC are prematurely activated due to loss of Dll4-mediated Notch signaling, but it is unclear if this is due to loss of spatial distribution of the proteins as suggested by the authors.

The reviewer is correct, complete ablation of DLL4 does not modulate spatial diversity. To test spatial regulation, one would have to flip regions from Dll4 high to Dll4 low and vice versa. This is not possible with current genetic approaches. As such we have modified the title to “Heterogeneous levels of Δ-like 4 Within a Multinucleated Niche Cell Maintain Muscle Stem Cell Diversity”.

To further address the relationship between heterogenous levels of Dll4 and stem cell diversity, we used a low dose TMX approach to reduce (not delete) Dll4 levels. The rationale is that we would retain spatial heterogeneity but with a compressed distribution of Dll4 levels. Compared to controls, we find reduced levels of Dll4 and Pax7 (Figure 4B, 4C), without a change in SC number (Figure 4A). This provides further evidence that SCs sit on a continuum based on Dll4 levels.

2. Mib regulates multiple Notch ligands. Does Mib1 deletion affects other Notch ligands, beyond Dll4, in myofibers?

It is widely accepted that *Mib1* genetic deletion abrogates the expression of all Notch ligands at the level of post-translational regulation (Koo et al., 2005, Koo et al., 2007). Therefore, we suggest that *Mib1* is regulating the Notch ligands that are expressed in muscle fibers. Unfortunately, the antibodies for Notch ligands other than DLL4 are either not available or not validated. Hence, it is not possible to analyze Notch ligand expression after *Mib1* deletion.

3. The authors state "Dll4 expression was also variable in regions devoid of SCs, suggesting the presence of a SC does not dictate Dll4 levels along muscle fibers". What is the frequency of Dll4 diversity independent of MuSC? Is it a rare event or is as frequent as the one correlated to the presence of MuSC on the myofiber?

A similar range of values and variability is observed in non-MuSC regions. We have stated this in the results.

4. While the authors found no correlation with the distance from the NMJ, this does not necessarily mean that there is no correlation with anatomically defined regions of the myofiber. in vivo, different regions of the myofiber are exposed to different local microenvironments, thus it cannot be rule out the possibility that such a correlation exists with other anatomical locations, that are lost upon myofiber isolation from the tissue.

Yes, we agree with the reviewer. In this manuscript, we focused on the NMJ and MTJ due to their well-defined positions and markers. We have re-written this to be more explicit: We find no consistent pattern of Dll4 protein along the fibers. Therefore, Dll4 spatial distribution does not map to these known anatomically defined regions of freshly-isolated single muscle fibers.

5. Distinction between Dll4 on myofiber and Dll4 captured by MuSC: it would be useful if the authors would include in the methods how they distinguish Dll4 on myofiber from the Dll4 captured by MuSC.

We appreciate the reviewer recommendation. The methods are now included in the manuscript.

6. In Figures 3I, 4K and S5J, it would be useful to include representative image of tissue sections, to support the quantification shown.

We have included representative images of Pax7 (SC) and laminin (basal lamina) stains in Figure 3—figure supplement 1.